# Filamin C dimerisation is regulated by HSPB7

Zihao Wang [1,2,3], Guodong Cao[1,2,24], Miranda P. Collier[1,2,24], Xingyu Qiu[1,2,24], Sophie Broadway-Stringer[4], Dominik Šaman[1,2], Jediael Z. Y. Ng [5], Navoneel Sen[1,2], Amar J. Azad [4,6], Charlotte Hooper[7], Johannes Zimmermann [8], Michael A. McDonough[9], Jürgen Brem [9,10], Patrick Rabe [9,11], Haigang Song [1,2], T. Reid Alderson [1,2,12,13], Christopher J. Schofield [9,14], Jani R. Bolla [15], Kristina Djinovic-Carugo[16,17], Dieter O. Fürst[18], Bettina Warscheid [8], Matteo T. Degiacomi [19,20], Timothy M. Allison[21], Georg K. A. Hochberg [5,22,23], Carol V. Robinson [1,2], Katja Gehmlich [4,7] ✉ & Justin L. P. Benesch [1,2] ✉

The biomechanical properties and responses of tissues underpin a variety important of physiological functions and pathologies. In striated muscle, the actin-binding protein filamin C (FLNC) is a key protein whose variants causative for a wide range of cardiomyopathies and musculoskeletal pathologies. FLNC is a multi-functional protein that interacts with a variety of partners, however, how it is regulated at the molecular level is not well understood. Here we investigate its interaction with HSPB7, a cardiac-specific molecular chaperone whose absence is embryonically lethal. We find that FLNC and HSPB7 interact in cardiac tissue under biomechanical stress, forming a strong hetero-dimer whose structure we solve by X-ray crystallography. Our quantitative analyses show that the hetero-dimer out-competes the FLNC homo-dimer interface, potentially acting to abrogate the ability of the protein to cross-link the actin cytoskeleton, and to enhance its diffusive mobility. We show that phosphorylation of FLNC at threonine 2677, located at the dimer interface and associated with cardiac stress, acts to favour the homo-dimer. Conversely, phosphorylation at tyrosine 2683, also at the dimer interface, has the opposite effect and shifts the equilibrium towards the hetero-dimer. Evolutionary analysis and ancestral sequence reconstruction reveals this interaction and its mechanisms of regulation to date around the time primitive hearts evolved in chordates. Our work therefore shows, structurally, how HSPB7 acts as a specific molecular chaperone that regulates FLNC dimerisation.

The beating of the heart applies continual cycles of biomechanical force to the contractile machinery and cytoskeletal support proteins that comprise cardiac muscle tissue. The molecular transitions that these proteins undergo, during both physiological and stress conditions, enable a variety of sensing and response behaviours, so-called mechano-signalling. The breakdown of, or aberrant, mechano-signalling pathways can lead to serious cardiac dysfunctions, including cardiomyopathies and heart failure[1].

A key protein within the cardiac cytoskeleton that is heavily involved in mechano-signalling is filamin C (FLNC; also known as γ-FLN, ABP-L, or FLN2). FLNC is found primarily in striated muscle cells, while the other two filamin paralogs, FLNB and FLNA, are expressed in all

---

other cell types[2]. FLNC has important cellular roles in signalling pathways, cell migration, differentiation and the assembly of the myofibrillar Z-discs[3,4], and contributes to rapid stabilisation and repair of myofibrillar damage[5]. Genetic variants in *FLNC* are associated with cardiomyopathy[6] and musculoskeletal myopathies[7], and its aberrant expression is related to cancer cell invasiveness[8,9].

FLNC's diverse roles apparently stem from it being both an actin-binding protein and a platform for binding a plethora of proteins ranging from membrane proteins to cytoskeletal network components, allowing it to integrate cellular architecture and signalling pathways[3]. The actin-binding domain of FLNC is located at its N-terminal end, and is followed by 24 immunoglobulin-like (Ig) domains (d1-d24). While d1-15 is thought to form a relatively elongated structure at rest, d16-24 appears to comprise an extendable mechanosensitive region that can bind to a variety of interaction partners[3]. The C-terminal domain, d24, is the site of homo-dimerisation of FLNC[10]. The dimer formation of FLNC is necessary for it to cross-link actin, and genetic variants in d24 that hinder dimerisation are associated with myofibrillar myopathy[11–13]. Numerous sites of phosphorylation have been identified on FLNC, in particular in d20, the hinge 2 region between d23 and d24, and d24. Those sites whose molecular consequences have been characterised appear to regulate their stability and interaction with different binding partners under stress conditions[14–16]. This nuanced link between phosphorylation events and the mechanical responses of FLNC adds a layer of regulatable adaptability to its role in cardiac mechano-signalling.

FLNC interacts with HSPB7, also known as cardiovascular heat-shock protein (cvHSP), via d24[17–20]. HSPB7 is a member of the small Heat-Shock Protein (sHSP) family, and is expressed specifically in heart and skeletal muscle[19,21]. The expression level of HSPB7 increases with age, and it is located in cardiomyocytes, either diffusely or at the Z-disc and the intercalated discs[18,22,23]. HSPB7 appears to play a crucial role in regulating the length of thin filaments during heart development and preventing the aggregation of monomeric actin[23]. Knockout, or missense and truncating variants, of HSPB7 lead to the formation of large aggregates of FLNC and a significant increase in actin filament bundles, which results in embryonically lethal cardiac defects in mice[17,18,23,24].

While less studied than most human sHSPs, HSPB7 appears to have features that are atypical of this family of molecular chaperones. Though it shares the common architecture of sHSPs (an α-crystallin domain, ACD, flanked by N- and C-terminal regions[25]), HSPB7 is unusual as it has been observed to be oligomerisation incompetent[26,27]. Nor HSPB7 does appear to display the canonical sHSP function of broadly inhibiting heat-induced protein aggregation[27,28]. These observations suggest that HSPB7 is not a promiscuous oligomeric chaperone, as in the archetypal sHSP, but instead may be specialised to a particular cellular role.

Here we test this hypothesis by examining the association between HSPB7 and FLNC from the tissue to the atomic levels. We validated the interaction between the two proteins in cardiac tissue and observed up-regulation and co-localisation of the two proteins in mouse models of biomechanical stress. Using a combination of structural approaches, we localise the binding between the two proteins to arise from a strong interaction between FLNC d24 and the HSPB7ACD, and solved the structure of the resulting heterodimer. We find that the equilibrium between FLNC homodimer and FLNC-HSPB7 heterodimer is modulated by phosphorylation of FLNC, and identified key amino-acid contacts that mediate this competitive binding. Our study reveals an ancient and uniquely specific sHSP:target interaction which appears to be a key regulatory mechanism for FLNC in its role of maintaining cytoskeletal integrity.

## Results
### Biomechanical stress leads to up-regulation and interaction of FLNC and HSPB7 in mouse hearts
To test the interaction between FLNC and HSPB7 in vivo, we used three mouse models of biomechanical stress and assayed the location and

abundance of each protein. The first model is a muscle LIM protein (MLP) knockout (KO) mouse. MLP is found exclusively in striated muscle, and its absence results in a distinctive tissue morphology that mirrors dilated cardiomyopathy and heart failure in humans[16]. Immunoprecipitation (IP) experiments using an anti-FLNC antibody followed by western blotting resulted in the detection of both FLNC and HSPB7 in the wild-type (WT) and MLP KO cardiac tissue (Fig. 1A). This demonstrates that these proteins interact in the mouse heart.

Noticeably more HSPB7 was pulled down from the MLP KO tissue. To determine whether this was due to increased expression under cardiac impairment, we evaluated levels of HSPB7 and FLNC in WT and MLP KO hearts. Western blots revealed significantly higher levels of both proteins in the MLP KO mice (Fig. 1B, left). We performed similar blots on tissue from mice that had been subject to biomechanical stress either due to transverse aortic constriction (TAC) surgery (which leads to pressure overload and induces heart failure) or chronic isoprenaline/epinephrine (IsoPE) treatment (which increases blood pressure and heart rate)[16]. In each case, we observed higher levels of FLNC and HSPB7 compared to the respective controls (Fig. 1B).

Next, we immuno-fluorescently stained frozen ventricular sections from the three mouse models for both HSPB7 and FLNC. In each model we found FLNC and HSPB7 to co-localise to varying degrees, and to do so primarily at the Z-discs and intercalated discs (Fig. 1C). In two models of biomechanical stress, namely MLP KO and TAC, co-localisation was quantitatively increased in the stress model compared to its respective control (WT or sham) – in the other it appears the tissue is already under stress even in the control so no increase is seen (Supplementary Fig. 1). Combined these results demonstrate that under biomechanical stress, FLNC and HSPB7 are up-regulated and interact with each other at sites of traction in cardiac tissue.

### HSPB7 exists as a monomer with limited anti-aggregation activity encoded by its N-terminal region
To investigate the nature of the interaction between FLNC and HSPB7, we turned to in vitro experiments on the purified proteins. We first expressed and purified the full-length HSPB7 (isoform 2, the longest of the three[19]) as well as two truncated constructs: one missing the N-terminal region (HSPB7$_{\Delta N}$, residues 78–175 of isoform 2) and one additionally missing the C-terminal region, i.e. just the core ACD (HSPB7$_{ACD}$, residues 78–162 of isoform 2) (Supplementary Fig. 2A). The N- and C- terminal regions are predicted to be disordered, and not to make substantive contacts with the ACD (Supplementary Fig. 2B). We obtained native mass spectrometry (MS) data for each construct at ~10 μM, and found that all three forms of HSPB7 were primarily monomeric (Supplementary Fig. 2C). This contrasts with equivalent experiments on other human sHSPs, which at micromolar concentrations are typically oligomeric in their full-length form, while their ACDs form dimers[16,29–31].

HSPB7 is considered ineffective in preventing the heat-induced aggregation of proteins, the canonical chaperone activity of other sHSPs[27]. However, one study showed some suppression of the aggregation of citrate synthase[27], so we used this target protein to perform a comparative assay of our three constructs. We found that full-length HSPB7 was able to suppress aggregation, albeit at high relative ratios (>1:1) of chaperone to target. However, neither HSPB7$_{\Delta N}$ nor HSPB7$_{ACD}$ displayed any anti-aggregation activity, even at ratios of 4:1 (Supplementary Fig. 2D). This shows that the N-terminal region encodes any canonical chaperone activity that HSPB7 has, in line with previous data[26,32].

### HSPB7 does not oligomerise due to the failure of its C-terminal region to bind the ACD
To identify the source of this unusual lack of oligomerisation by HSPB7, we examined the regions of its sequence that mediate assembly in other sHSPs. Typically, sHSPs oligomerise via an interaction

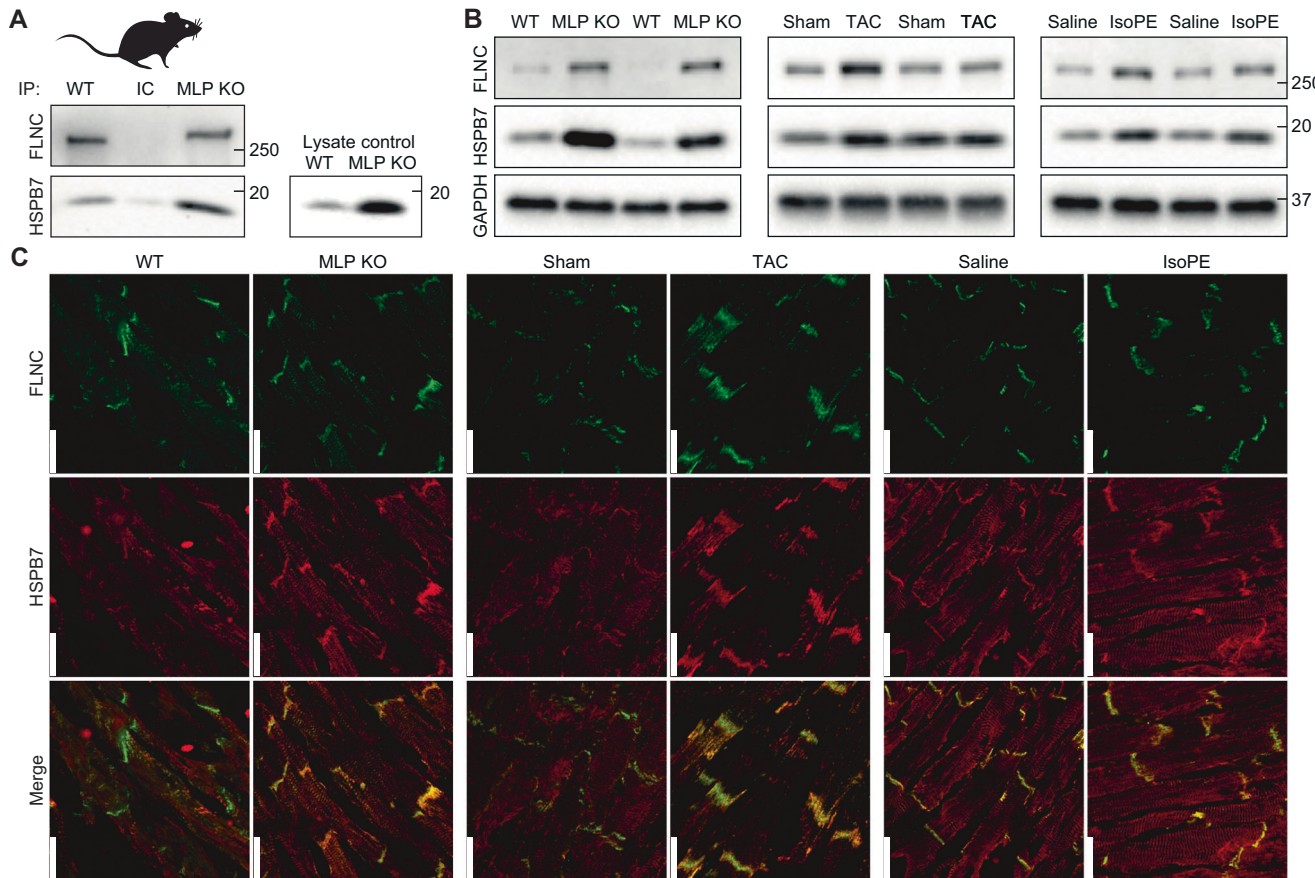

**Fig. 1 | HSPB7 and FLNC are up-regulated and interact in biomechanical stress models in mouse. A** Co-immunoprecipitation of HSPB7 and FLNC from MLP KO mouse ventricular tissue. The co-precipitation of the two proteins when pre-cipitating FLNC suggests they form (at least part of) a complex. An isotype control, where a non-specific antibody (rather than for FLNC) raised in the same species was used as a negative control and showed no bands, whereas a positive control using an aliquot of lysate showed clear bands for HSPB7. Molecular mass bands are marked, in units of kDa. One biological repeat, immunoprecipitation experiment performed in triplicate, one representative shown. **B** Western blots for FLNC and HSPB7 in MLP KO, TAC and IsoPE mouse ventricular tissue. GAPDH was used as the

loading control. Sham and saline are the respective specific negative controls for TAC and IsoPE. In all stress treatments, HSPB7 and FLNC are observed at higher levels. Molecular mass bands are marked, in units of kDa. Two biological replicates (i.e. two mice), blots repeated twice, one representative blot shown.
**C** Immunofluorescence of HSPB7 and FLNC from frozen sections of ventricular tissues from TAC, IsoPE and MLP KO mouse models. Sections were stained for FLNC (Green) and HSPB7 (Red). In all stress treatments, the two proteins are up-regulated and co-localise. For quantification of co-localisation see Supplementary Fig. 1. Scale bars are 25 μm. Two biological replicates (i.e. two mice), two images taken from each sample, one representative image shown.

mediated by an "IXI motif", where X refers to a variable amino acid, and the I refers typically (but not exclusively) to Ile[33]; in human sHSPs, the motif is generally manifested as IPI/V. The IXI motif in the C-terminal region of one monomer facilitates assembly by binding into the ACD β4-β8 groove of a neighbouring monomer[30,33]. HSPB7 contains the sequence IKI at its extreme C-terminus. To test whether this can bind to the ACD, we incubated a peptide mimicking the last eight residues of the C-terminus (residues 168–175, TFRTEIKI) with $HSPB7_{ACD}$ and performed native MS experiments. We observed no significant inter-action between the two (Supplementary Fig. 3). Control experiments under the same conditions for the ACDs of HSPB1 and HSPB5 (human sHSPs that oligomerise[33]) showed binding of their cognate peptides with ≤100 μM $K_D$, consistent with previous reports[29,30].

To test whether this absence of binding was due to the IXI motif or the $HSPB7_{ACD}$ β4-β8 groove, we performed experiments in which we incubated either $HSPB1_{ACD}$ or $HSPB5_{ACD}$ with the HSPB7 peptide, and $HSPB7_{ACD}$ with either the HSPB1 or HSPB5 peptides. We found that neither of the ACDs were able to bind the HSPB7 peptide to a sig-nificant extent, whereas HSPB7 was able to bind both HSPB1 and HSPB5 peptides (Supplementary Fig. 3). These observations indicate that the failure of HSPB7 to oligomerise is due to its C-terminal region being different to that of other human sHSPs, either because it does not

contain the IXI motif in the IPI/V form found in the other human sHSPs, or due to the IXI motif being at the extreme end of the sequence and hence lacking flanking residues that provide additional stabilisation of the peptide-ACD interaction[30,34].

## The HSPB7 structure reveals its failure to dimerise is due to a destabilised pairing of β6 + 7 strands

To investigate why none of the HSPB7 constructs dimerise to an appreciable extent, we turned to X-ray crystallography. We under-took crystallisation trials on $HSPB7_{ACD}$ and $HSPB7_{ΔN}$, but did not obtain any protein crystals. We speculated that this might be due to a solvent-accessible cysteine (C131) leading to non-specific inter-molecular interactions during crystallisation, and hence generated the C131S mutant of both constructs. Of these, $HSPB7_{ACD}{}^{C131S}$ returned good crystals, which diffracted to 2.2 Å resolution; the structure was solved by molecular replacement in the $P4_12_12$ space group (Supplementary Table 1). The fold of the protein is Ig-like, with six β-strands packed into a β-sandwich arrangement very similar to that observed for other sHSP ACDs[35,36] (Fig. 2). Each asymmetric unit contains three monomer chains; two of the three monomers are well resolved, while the third displays disorder in the amino acids that comprise the loop linking the β5 and β6 + 7 strands (residues 112–118,

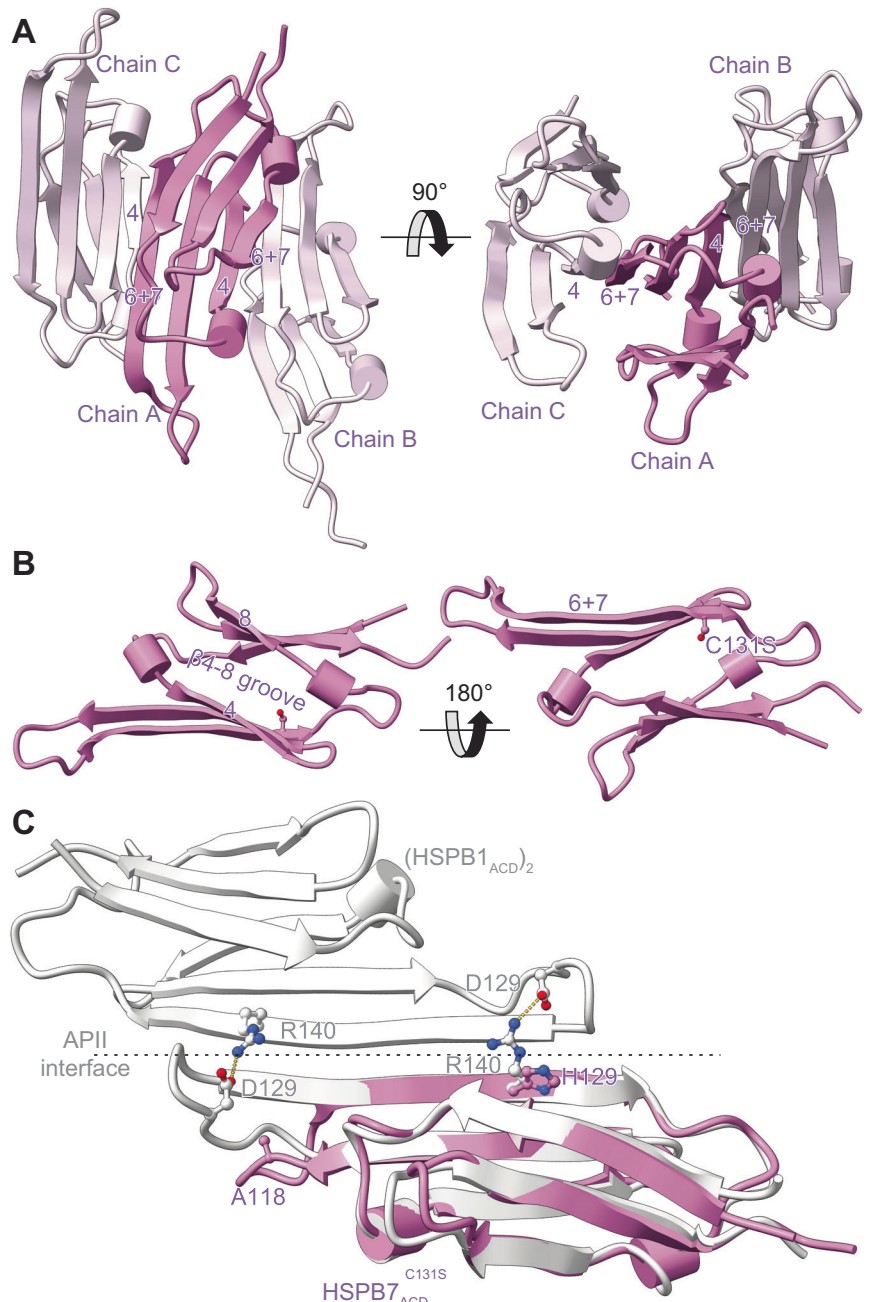

**Fig. 2 | Crystal structure of the HSPB7 monomeric ACD. A** The asymmetric unit of HSPB7$_{ACD}$$^{C131S}$ contains three chains (chains A–C), packed through the neighbouring β4 and β6 + 7 strands. **B** The Ig-fold is very similar to that of other sHSP ACDs: it contains six β strands, with a groove between β4 and β8 and an extended "β6 + 7" strand. **C** Overlay of a HSPB7$_{ACD}$$^{C131S}$ (purple) structural view with one of the closely related HSPB1 (white) reveals how the absence in HSPB7 of specific salt bridges (e.g. D129-R140 in HSPB1, highlighted here) that are conserved in the other HSPBs (Supplementary Fig. 10B) have weakened the dimer interface such that monomers are the dominant species in solution.

LAADGTV), a phenomenon that has also been observed in other sHSPs upon monomerisation[37].

The monomers within the asymmetric unit contact each other through crystallographic interfaces between neighbouring anti-parallel β4 and β6 + 7 strands that form long chains of monomers in the crystal. The lack of a head-to-head dimer in the crystal form is different to most sHSP ACD structures that have been determined, but is consistent with our native MS data that showed monomers to be the dominant form in solution. A rationale for this might be that the β6 + 7 strand is shorter in HSPB7 compared to that found in the canonical human sHSPs; in particular it is missing charged amino acids at positions 118 and 129 that form inter-monomer salt bridges in HSPB1

and HSPB5 (Fig. 2C). These differences likely act to reduce the strength of a putative anti-parallel interaction of two β6 + 7 strands (as per the canonical metazoan sHSP dimer), with the result that HSPB7 exists primarily as a monomer.

## HSPB7 binds to FLNC d24 strongly by out-competing homo-dimerisation

Having developed an understanding of the HSPB7 structure, we proceeded by expressing and purifying domain 24 of FLNC (FLNC$_{d24}$), which forms the dimerisation interface of FLNC and has been identified as the interaction site for HSPB7[12,17]. Native MS measurements of the FLNC$_{d24}$ construct revealed it to be dimeric, in line with the

literature[10]. We then incubated it at a 1:1 ratio with each of HSPB7, HSPB7$_{\Delta N}$ and HSPB7$_{ACD}$ in turn, and observed the appearance of new charge state series that could be assigned unambiguously and exclusively to heterodimers of FLNC$_{d24}$ and the HSPB7 construct (Fig. 3A, Supplementary Table 2). These heterodimers were formed by HSPB7$_{ACD}$ to the same abundance as in the other constructs, revealing the interaction to be mediated by the ACD of the sHSP.

We assayed the strength of this interaction quantitatively for HSPB7$_{\Delta N}$ and HSPB7$_{ACD}$ by performing experiments at a range of FLNC$_{d24}$ concentrations; the instability of full-length HSPB7 (some unfolding is observed in the native MS spectra, and we found the protein could readily precipitate in vitro) precluded reliable equivalent experiments for it. This allowed us to obtain dissociation constants ($K_D$) for the FLNC$_{d24}$:HSPB7$_{\Delta N}$ heterodimer as $6.7 \pm 1.4$ nM and FLNC$_{d24}$:HSPB7$_{ACD}$ heterodimer as $3.9 \pm 0.9$ nM, significantly tighter binding than we measured for the FLNC$_{d24}$ homodimer ($75.9 \pm 8.4$ nM), by >6 kJmol$^{-1}$ (see later, Fig. 4). Control experiments using the ACD of HSPB1 and HSPB5 displayed negligible binding to FLNC$_{d24}$ (Supplementary Fig. 4). These results show that FLNC and HSPB7 form a strong, specific heterodimer through the interaction of FLNC$_{d24}$ with the ACD of HSPB7.

### The structure of the HSPB7:FLNC heterodimer reveals the molecular mechanism of competition

Given the nature of the heterodimer between HSPB7$_{ACD}$ and FLNC$_{d24}$ as composed of two well-structured units interacting with nanomolar affinity, we considered them a strong candidate for co-crystallisation. We were successful in obtaining crystals of the complex, which diffracted to 2.9 Å, and solved the structure in the $P3_221$ space group (Supplementary Table 1). The asymmetric unit comprised a single heterodimer, with the interface centred around a parallel pairing of FLNC$_{d24}$ strand C with HSPB7$_{ACD}^{C131S}$ β4, and the donation of sidechains from each strand into an adjacent hydrophobic groove on each monomer (β4-β8 groove on HSPB7$_{ACD}$, and C-D groove on FLNC$_{d24}$) (Fig. 3B). Note that this is a different site and mode of association to that observed in the dimerisation of sHSP ACDs, which occurs via antiparallel pairing of β6 + 7 strands. Specifically, the HSPB7 β4-β8 groove accepts M2667 and M2669 from FLNC, while the groove between strands C and D accepts I102. The interface is thus mainly comprised of hydrophobic contacts made between the monomers' sidechains (Fig. 3C), with additional stability conferred by polar contacts between the chains (Supplementary Table 3).

To validate the interface observed crystallographically, we performed hydrogen-deuterium exchange (HDX)-MS experiments, whereby the solvent accessibility of backbone amides in proteins is probed in solution[38]. We obtained excellent sequence coverage for both HSPB7$_{ACD}$ (97%) and FLNC$_{d24}$ (98%), allowing us to compare the uptake of deuterium across the sequence of both proteins when they were labelled in complex versus alone. The rate of deuterium uptake of several peptide products in each of HSPB7$_{ACD}$ and FLNC$_{d24}$ was slowed when they were co-incubated, consistent with them forming a heterodimer whose interface protects interacting regions from exchange (Fig. 3D–F). For FLNC$_{d24}$, the most protected region at all time-points is residues 2658–2682, which spans from the middle of strand B to the start of strand D (Fig. 3D). For HSPB7$_{ACD}$, the most protected region encompasses residues 101–117, incorporating both β4 and β5 (Fig. 3D). These results are in close accord with the dimer interface we observed in the crystal structure, including with respect to the pairing of strands C and β4.

Bearing in mind that HDX-MS operates at a resolution defined by the enzymatic digestion (and hence unaffected residues can appear protected simply by being within the same peptide product as residues that truly are), mapping the protection from deuteration measured onto our structure emphasises how protection is highest near the crystallographically observed interface (Fig. 3F). This also shows that

the C131S mutation that we introduced to aid crystallisation, but was absent in the construct used in HDX-MS, does not alter the location of the dimer interface. The protection measured between HSPB7 β3 and β4 (residues 90–100) can be rationalised by the salt bridge formed between D100 and FLNC H2686. The protection at the C-terminal end of HSPB7$_{ACD}$, from the β8-9 loop to the C-terminus (residues 147–162) appears to arise from an interaction between the loop and the Gly-Ser residue pair at the extreme N-terminus of our FLNC$_{d24}$ construct that are residual from its production as a His-tagged protein, and are likely artefactual. Together, our crystallography and HDX-MS data provide a structure, validated in solution, that explains the tight binding of FLNC d24 to the ACD of HSPB7.

### Phosphorylation of FLNC domain 24 regulates its homo- and hetero-dimerisation

Numerous phosphorylation sites have been discovered on FLNC, many of which become modified upon cellular stress of various types. Examining these in light of the overlapping homo- and hetero-dimerisation interfaces of FLNC$_{d24}$ revealed two sites for further interrogation, pT2677 and pY2683. The former has been reported in studies examining the regulation of phosphorylation in infarct hearts[39], and Covid-19 infection[40] (a disease in which there is significant cardiovascular involvement[41], including stress cardiomyopathy[42]). pY2683 has been reported in studies examining exercise-induced signalling[43,44], or phosphorylation in a variety of tumour tissues, including bone cancer, osteosarcoma and glioma[45]. Based on manual analysis of tandem-MS spectra from the deposited datasets, we were able to validate the presence of both phosphorylation sites (Supplementary Fig. 5).

For technical reasons, we were not able to detect the equivalent phosphorylations in our mouse models of cardiac biomechanical stress. We think this is due primarily to the sample quantities required, which would not be in keeping with ethical use of animals in research. Therefore, while the presence of these phosphorylations in human muscle remains compelling, we also consider modifications at T2677 (on the edge of the interface) and Y2683 (in the middle) as being contrasting levers for assessing the biophysical effect of perturbation of the dimer interface we identified in our structural work. To mimic the effect of phosphorylation at each of these sites, we expressed and purified the FLNC$_{d24}$ T2677D and Y2683E mutants for in vitro experiments. We chose to make the mutations T → D and Y → E to mimic pT and pY because these were the best match given the respective alpha-carbon to oxygen distances (i.e. D < E, pT < pY from D: 3.3 Å, pT: 3.7 Å, E: 4.6 Å, pY: 7.7 Å; average of two Cα-O distances in each case).

Native MS data for each of the phosphor-mimicking mutants and the WT FLNC$_{d24}$ at the same protein concentration showed differing amounts of monomer and dimer in each (Fig. 4A). Compared to the WT, FLNC$_{d24}^{T2677D}$ showed an increase in abundance of homodimer, while FLNC$_{d24}^{Y2683E}$ showed a decrease. From titration experiments we obtained the corresponding $K_D$s as $75.9 \pm 8.4$ nM (WT), $8.5 \pm 1.0$ nM (T2677D) and $4047 \pm 557.0$ nM (Y2683E) (Fig. 4B). These data suggest that phosphorylation of each site differentially regulates the homo-dimerisation of FLNC, with T2677 stabilising the dimer (by 5.4 kJmol$^{-1}$) and Y2683 weakening it (by 9.9 kJmol$^{-1}$) (Fig. 4C).

We next performed similar experiments for the heterodimer. We found that the interaction with HSPB7$_{ACD}$ is even stronger for FLNC$_{d24}^{Y2683E}$ ($2.2 \pm 0.6$ nM) than the WT ($3.9 \pm 0.9$ nM), a stabilisation of 1.5 kJmol$^{-1}$ (Fig. 4C). However, the opposite is true for FLNC$_{d24}^{T2677D}$ ($14.1 \pm 0.5$ nM), which is destabilised relative to the WT by 3.2 kJmol$^{-1}$. Indeed, together with the stabilisation conferred by the phospho-mimic to the homodimer, this means that FLNC$_{d24}^{T2677D}$ actually favours homo- versus hetero-dimerisation (Fig. 4C).

To validate these findings, we obtained comparative HDX-MS data for the phosphomimics and WT FLNC$_{d24}$ in complex with HSPB7$_{ACD}$. Focusing on HSPB7$_{ACD}$ only (because FLNC$_{d24}$ has a different sequence

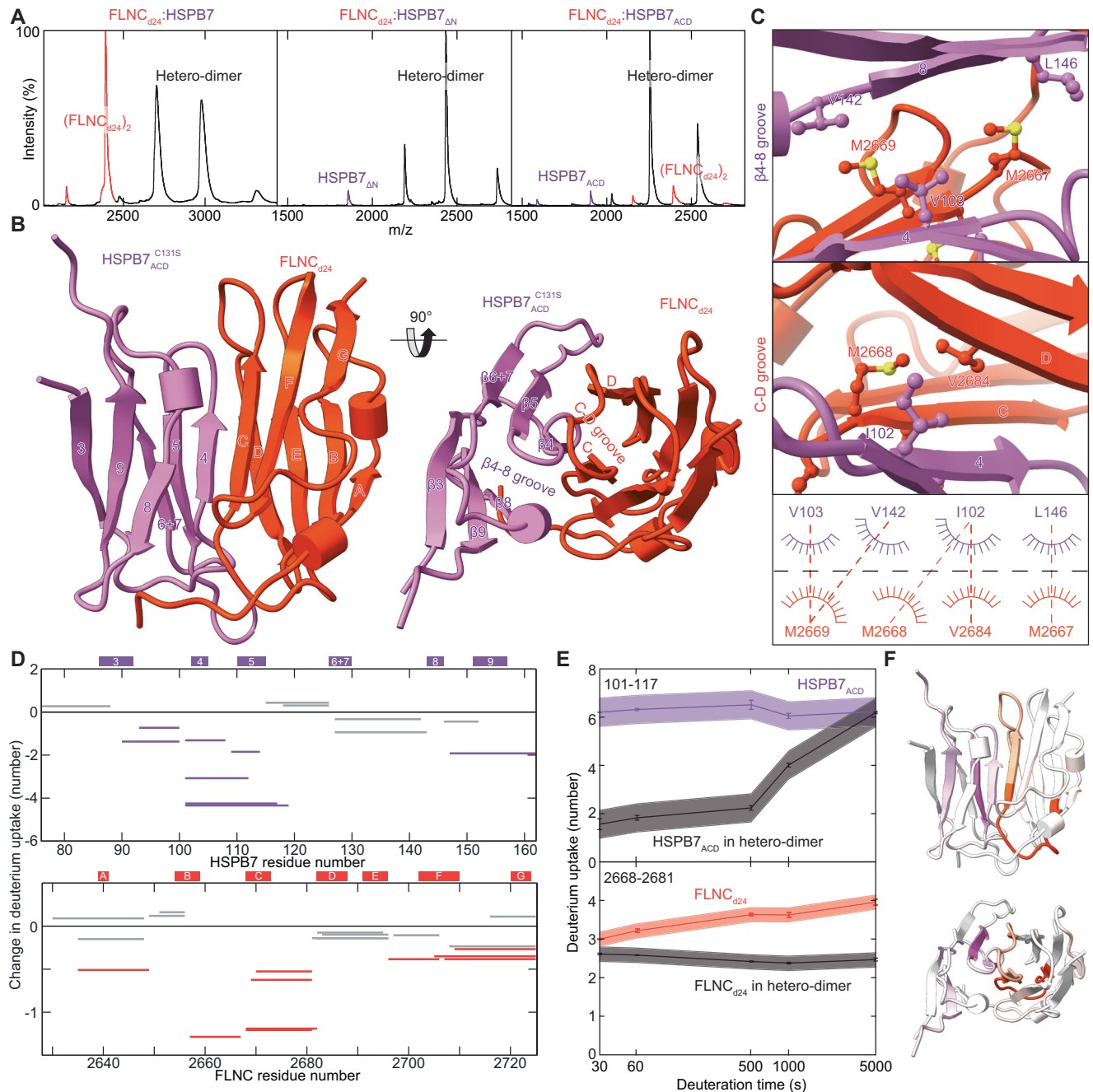

**Fig. 3 | Hetero-dimerisation of HSPB7 and FLNC. A** Native MS experiments show that FLNC$_{d24}$ (red) forms heterodimers (black) with HSPB7, HSPB7$_{\Delta N}$ and HSPB7$_{ACD}$ (purple). No complexes between the FLNC$_{d24}$ homodimer and HSPB7 are observed, revealing how homo- and hetero-dimerisation are directly competitive. Proteins were mixed at a monomer ratio of 1:1, but the instability of full-length HSPB7 means it was somewhat depleted in solution such that FLNC$_{d24}$ was effectively in excess (and FLNC$_{d24}$ is visible in the spectrum). **B** The FLNC$_{d24}$:HSPB7$_{ACD}$$^{C131S}$ heterodimer structure reveals an interface that involves the HSPB7 β4, β8 strands and FLNC$_{d24}$ strands C and D, centred on the parallel pairing of strands C and β4. **C** The hydrophobic β4-8 groove on HSPB7 accepts M2667 and M2669 from FLNC (top), while the groove between strands C and D on FLNC$_{d24}$ accepts I102 from HSPB7 (middle). This leads to a network of hydrophobic interactions that stabilise the heterodimer (bottom). **D** HDX-MS analysis of HSPB7$_{ACD}$ and FLNC$_{d24}$ in the heterodimer compared to in isolation. Woods plots showing the difference in

deuterium uptake of HSPB7$_{ACD}$ (top) and FLNC$_{d24}$ (bottom) in the two states, at a labelling time of 500 s. The y-axis is calculated as the uptake for the isolated proteins minus that in the heterodimer; negative values denote protection from exchange in the heterodimer. Three technical repeats were carried out, and peptides were considered significantly different if $p < 0.01$ (coloured: orange or purple). **E** Plot of deuterium uptake versus exposure time of the most protected peptide in HSPB7$_{ACD}$ (upper) and FLNC$_{d24}$ (lower) in the heterodimer compared to this same peptide in the isolated protein. Because HSPB7 is monomeric, in the absence of FLNC, this peptide is solvent-exposed and therefore has consistently high uptake values. Error bars refer to the standard deviation of three repeats at each time-point, and the shaded bands 99% confidence. **F** Mapping the uptake difference at 500 s (as a fraction of the theoretical maximum) onto the heterodimer protein structure shows that protection is centred on the dimer interface that we found in our crystal structure.

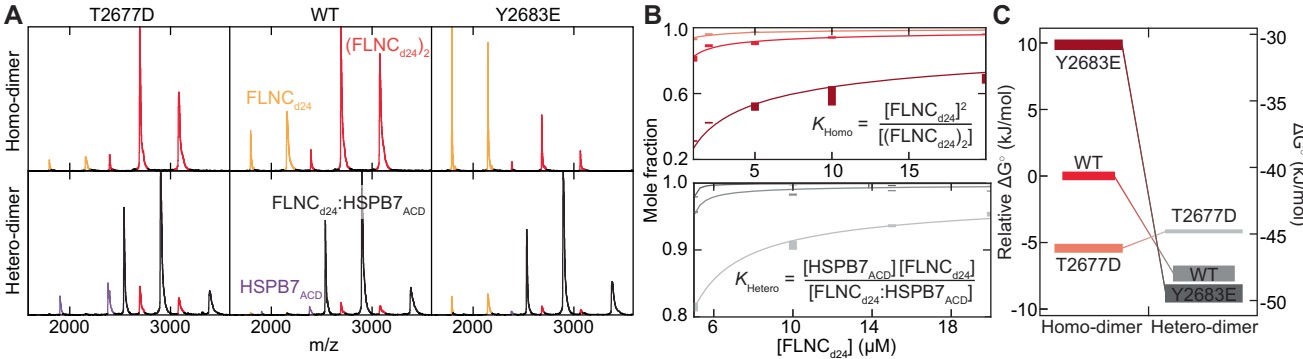

**Fig. 4 | Mimicking phosphorylation at T2677 and Y2683 has opposing effects in modulating FLNC_{d24} homo- and hetero-dimerisation. A** Native mass spectra at 1 μM of FLNC_{d24} WT (middle) and the phosphomimics T2677D (left) and Y2683E (right) show differences in the relative abundances of the monomer (yellow) and dimer (red) (upper row). T2677D forms a stabilised homodimer, relative to WT; Y2683 a destabilised dimer. Native MS spectra at 5 μM of each of HSPB7_{ACD} and either FLNC_{d24} WT or one of the phosphomimics (lower row). Heterodimers are the dominant species in each of the spectra, but a notable abundance of FLNC (red, and

yellow) and HSPB7_{ACD} (purple) can be observed in the case of the T2677D mutant, and to a lesser extent for WT and Y2683E. **B** Titration curves for homo- (upper) and hetero-dimerisation (lower). WT: red and medium grey; T2677D: salmon and light grey; Y2683D: burgundy and dark grey. Error bars represent ±1 standard deviation of the mean (*n* = 3). **C** Free energy diagram showing the differences in stability for homo- and heterodimers in each of the three forms of FLNC. The vertical thickness of the bars corresponds to the error in our estimates. Lower free energies correspond to higher stability. See also Supplementary Methods.

in the three constructs and hence generates different, and thus not directly comparable, peptides) we found that the peptide diagnostic of the dimer interface (residues 101–117; incorporating both β4 and β5, Fig. 3D, E) showed decreased protection from deuteration for FLNC_{d24}^{T2677D} versus WT, and increased protection of FLNC_{d24}^{Y2683E} (Supplementary Fig. 6). This is consistent with the order of interface strengths that we observed in the native MS experiments, and underscores how phosphorylation at these two residues can regulate the interaction with HSPB7.

## Molecular dynamics analysis reveals the residue contacts altered by phosphorylation

To isolate the amino acid contacts that underpin these phosphorylation-induced changes in FLNC homo- and hetero-dimerisation we turned to molecular dynamics (MD) simulations. We first generated a T2677- or Y2683-phosphorylated FLNC_{d24} homo-dimer structure (FLNC_{d24}^{pT2677}, FLNC_{d24}^{pY2683}), by adding a doubly charged phosphate group to the sidechains of either T2677 or Y2683 in the WT FLNC_{d24} structure (Fig. 5A). We also did the same for the heterodimers, generating FLNC_{d24}^{pT2677}:HSPB7_{ACD} and FLNC_{d24}^{pY2683}:HSPB7_{ACD} from our FLNC_{d24}:HSPB7_{ACD} structure.

We ran 1-μs MD simulations under the same conditions for all six structures. All remained stable over the trajectories (Supplementary Fig. 7A), allowing us to interrogate them to understand how they differ in the inter-monomer interactions due to the introduction of the charged phosphate groups (Supplementary Data 1: occupancy of all hydrogen bonds). Examining the homodimers first (Fig. 5B, Supplementary Figs. 8 and 9), and comparing each mutant individually with the wild type, we found that pT2677 forms a salt bridge with R2690 on the opposite monomer. This likely strengthens the dimer interface in the phosphorylated form. In contrast, while the WT Y2683 is able to form an intermolecular hydrogen bond with E2681 across the dimer interface, pY2683 does not. Instead, the negative charges on the phosphate group reside within a predominantly negatively charged environment, and are coulombically repelled by the surrounding residues. These changes combine to destabilise the homodimer interface of FLNC_{d24}^{pY2683} relative to the WT.

Examining the heterodimers next (Fig. 5C, Supplementary Figs. 8 and 9), we noted that phosphorylation of T2677 leads to the loss of a hydrogen bond between the FLNC G2674 backbone carbonyl and the HSPB7 N107 sidechain (Fig. 5C, upper). We also see strengthened interaction between HSPB7 T143 (on β8) and FLNC D2712 (in the loop

between strands F and G) in pT2677. The increase in occupancy of this hydrogen bond is, however, insufficient to offset the destabilisation of the heterodimer caused by the loss of the G2674-N107 bond. In the pY2683 heterodimer, the phosphorylated pY2683 binds with HSPB7_{ACD} R113 via a salt bridge, which is not formed in the WT (Fig. 5C, lower). We also find that E99 from HSPB7 forms bonds with H2686, N2689 and R2690 in the phosphorylated form, further stabilising the dimer interface. We note that performing equivalent MD simulations of phosphomimics return quantitatively very similar results to those obtained for phosphorylated forms (Supplementary Fig. 7B). These results therefore not only justify the use of phosphomimics in our experimental work described earlier, but also provide residue-specific detail for the phosphorylation-mediated regulation of stability of FLNC homo- and hetero-dimerisation (Fig. 4).

## FLNC:HSPB7 hetero-dimerisation is an ancient, regulated interaction

Given the apparently strong, specific and regulatable interaction between FLNC_{d24} and HSPB7, we were interested in examining the evolutionary conservation of this complex. Therefore, we constructed phylogenetic trees of HSPB7, from which we could trace it into organisms as distantly related to humans as the tunicate *Styela* (a jawless chordate which possesses a heart), but not in the cephalochordate *Branchiostoma* (Supplementary Fig. 10A). Interestingly, when examining the set of 113 HSPB7 sequences, the residues that we identified as key in acting to prevent dimerisation of human HSPB7 (A118 and H129) are highly conserved across all HSPB7s, while in the other sHSPs they are highly conserved as Asp and Arg, respectively (Supplementary Fig. 10B). This strongly suggests that HSPB7 is predominantly monomeric in all organisms in which it is found.

To assess the interaction between FLNC and HSPB7, we designed an in silico competitive-binding assay (Supplementary Fig. 10C-D). In our approach, we supplied AlphaFold two chains of both FLNC_{d24} and HSPB7, such that it had the opportunity to either assemble the proteins into hetero- or homodimers. For the human pair, this approach identified hetero-dimerisation as favoured, and returned a model that matched our crystal structure closely. A negative control experiment, using HSPB5 (which our experiments show does not bind to FLNC_{d24}, Supplementary Fig. 4), led to homodimers of it and FLNC instead.

The alignment of these results with our experiments gave us confidence in our computational approach, and we thus inferred the dimerisation mode for various FLNC_{d24}:HSPB7 pairs along this

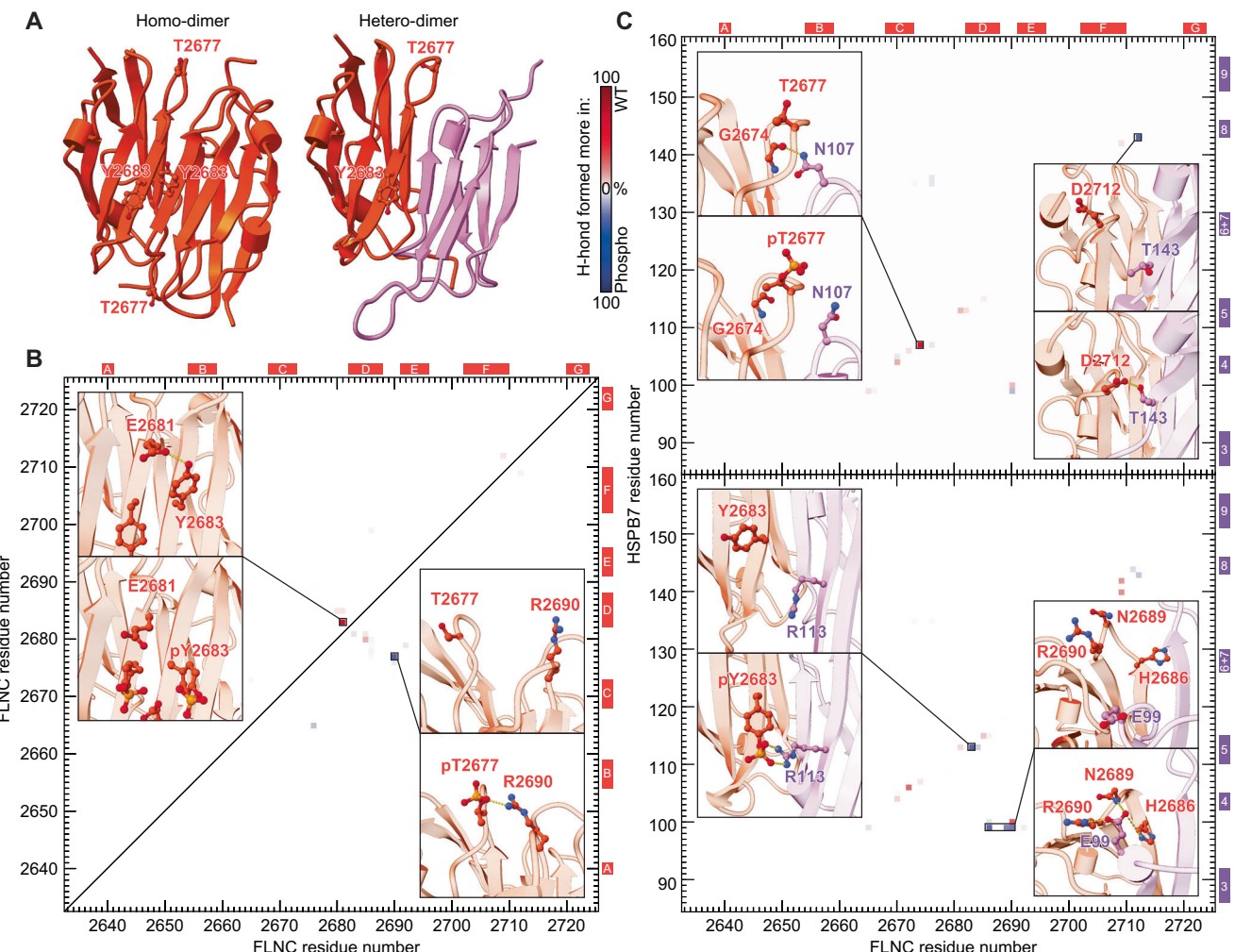

**Fig. 5 | MD simulations reveal how phosphorylation modulates the strength of the FLNC_d24 homo- and heterodimer interfaces. A** The location of the T2677 and Y2683 phosphorylation sites on the FLNC_d24 homodimer (PDB 1V05) and FLNC_d24:HSPB7_ACD^C131S heterodimer structures is near the interface. FLNC: orange; HSPB7: purple. **B** Difference map of hydrogen bond occupancy between WT and either pT2677 (lower right triangle) or pY2683 FLNC_d24 (upper left) homodimers. Each pixel corresponds to a particular inter-monomer contact between residue pairs. Blue: hydrogen bond formed more in the phosphorylated form; red: hydrogen bond formed more in the WT. A highly occupied hydrogen bond formed across the WT homodimer interface (between E2681 and Y2683) is lost upon phosphorylation of Y2683 (WT 71%; pY2683 0%). And a new interaction with R2690 is formed upon phosphorylation of T2677 (WT 0%; pT2677 26%) (insets).

**C** Difference maps of hydrogen bond occupancy between WT and pT2677 (upper) or pY2683 FLNC_d24:HSPB7_ACD^C131S heterodimers (lower). Colouring as in (**B**). The most substantial differences are highlighted with representative frames from the trajectories showing the contacts made (insets). These are: a highly occupied hydrogen bond formed across the WT heterodimer interface (between G2674 and N107) lost upon phosphorylation of T2677 (WT 63%; pT2677 1%); and a new interaction is formed between pY2683 with R113 that is absent in the WT (WT 0%; pY2683 32%). Altered interactions are also observed between residues that do not directly involve the phosphorylated site, e.g. a contact between D2712 and T143 (WT 10%; pT2677 38%) and ones between the cluster H2686/N2689/R2690 and E99 (WT 0/0/22%; pY2683 28/41/45%). For a full inventory of hydrogen bond occupancies, see the Supplementary Data 1.

phylogenetic tree. We found that hetero-dimerisation was preferred in extant organisms as distantly related to humans as *C. milli* (elephant shark; in the class chondrichthyes). Moreover, the *C. milli* heterodimer model is extremely similar to the human heterodimer (Supplementary Fig. 10). To estimate when this heterodimer first emerged, we used ancestral sequence reconstruction to resurrect the HSPB7 sequences in the (long since extinct) ancestors of humans and *C. milli*, and humans and the tunicate *Styela*. AlphaFold also returns heterodimer models of these ancestral HSPB7s (with either human or *C. milli* FLNC_d24) of equivalent structure to the extant forms (Supplementary Fig. 10). These results suggest that HSPB7's interaction with FLNC likely evolved along the branch leading to the last common ancestor of tunicates and humans. This animal was a chordate that likely had a primitive tubular heart[46], in line with HSPB7's present-day function in humans.

Having elucidated that the interaction between HSPB7 and FLNC is >400 million years old, we asked whether there is evidence for the

regulatory fine-tuning we uncovered for T2677 and Y2683 to be equally ancient. We noted that both sites also exist in the sequence of FLNC_d24 from *C. milli* and *Styela*, suggesting this to be the case. We also examined the sequence of FLNC_d24 in archetypal organisms across the relevant diversity of organisms, including reptiles, amphibians, fish and birds. In all cases, the two sites were conserved, suggesting that they are available for phosphorylation (assuming the presence of an appropriate kinase[47]). This evolutionary analysis therefore implies that not only is the interaction between HSPB7 and FLNC long-established, but so too is its regulation by post-translational modification during stress-signalling.

## Discussion
### FLNC dimerisation and hence diffusional mobility is regulated by HSPB7
This study reveals that FLNC and HSPB7 interact in murine cardiac tissue, with both proteins being up-regulated under biomechanical

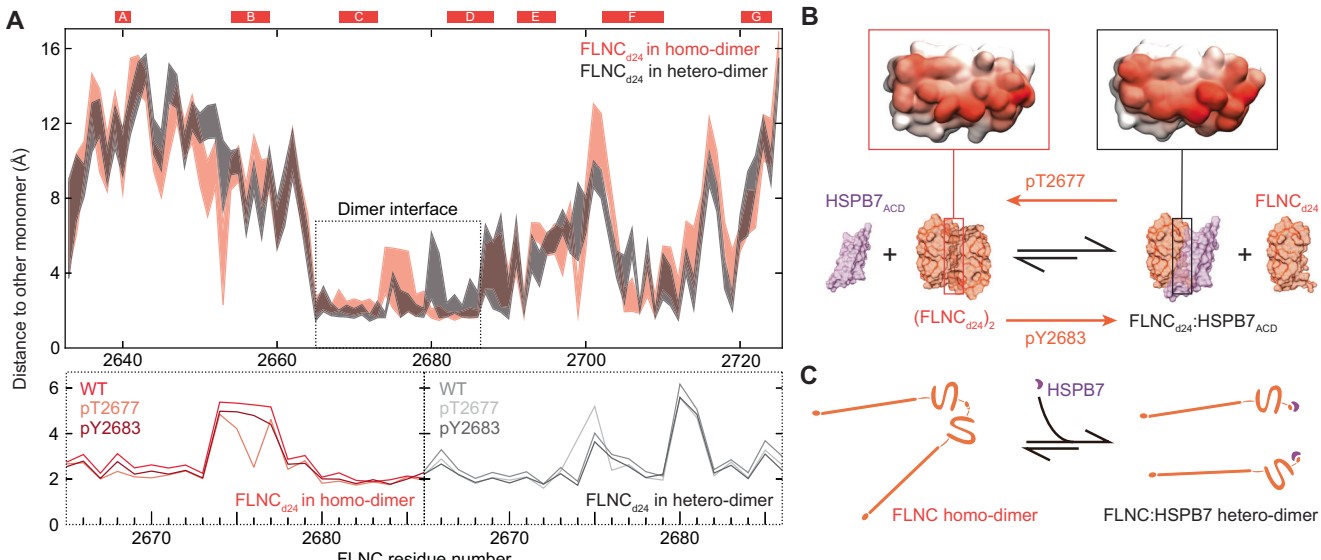

**Fig. 6 | Overlapping interfaces for homo- and hetero-dimerisation lead to an equilibrium that can be regulated by phosphorylation to adjust FLNC mobility. A** Plot of the distances between subunits in the homo- (orange) and heterodimer (black), as a function of sequence position in our MD simulations. The bands are defined by the closest (lower value) and average (upper value) distance over the length of the simulation (upper). The two bands overlap well, demonstrating the overall similarity of the interfaces in the two dimers. The core dimer interface is marked (dashed box), and expanded in the lower panel, which shows the average distance only, but for each of the WT, pT2677 and pY2683 simulations. These graphs show that the dimer interface distances are highly similar across the three proteins, with minor differences found only near pT2677 (for that protein). This is consistent with the overall structure of the dimer interface being largely similar,

with the major differences between WT and phosphorylated forms lying in the hydrogen-bonds across the interface (Fig. 5). **B** Colouring residues on the surface in proportion to the closest distances of a WT FLNC monomer to its counterpart reinforces the similarity in binding site between homo- and heterodimer (insets). This allows us a simple view of the equilibria involved, where $FLNC_{d24}$ homo- and hetero-dimerisation are competitive. The equilibrium favours the heterodimer for WT FLNC, and can be shifted in either direction by phosphorylation. **C** The shifting of the equilibrium effectively adjusts the state of FLNC, either stabilising it in the actin-crosslinking-competent homodimer form, or effectively releasing it from this role by generating the much smaller heterodimer, which is likely much more mobile in the cell.

stress in vivo. This conclusion is consistent with previous studies that have found that FLNC and HSPB7 can co-localise and interact in cells[17,18]. The interaction between FLNC and HSPB7 has previously been shown to occur at d24 of FLNC[17]; our results confirm this and map the binding of $FLNC_{d24}$ to the ACD of an HSPB7 monomer, in line with computational predictions[20]. Close examination of the $FLNC_{d24}$:$HSPB7_{ACD}^{C131S}$ heterodimer structure with the published structure of the $FLNC_{d24}$ homodimer (PDBID:1V05)[10] shows that the interface surfaces overlap very closely, and thereby provides a structural rationale for the competitive binding between the two proteins (Fig. 6A, B). Notably, our experimental data show that the heterodimer has a low nM $K_D$, significantly stronger than the FLNC homodimer. Taken together, these results can be viewed as HSPB7 effectively acting to shift the equilibrium between the FLNC dimer and monomer towards dissociation, with the dissociated FLNC monomer tightly capped by HSPB7 (Fig. 6C).

The full-length FLNC:HSPB7 heterodimer (26 domains, ~310 kDa) is much smaller than the FLNC homodimer (50 domains, ~582 kDa). This difference in size leads to a larger diffusion coefficient for the free heterodimer, with the result that FLNC is more mobile in solution in the presence of HSPB7. A consequence of up-regulation of HSPB7 during biomechanical stress might therefore be to increase the mobility of FLNC, such that it translocates more rapidly to force-induced microlesions in the sarcomeric structure, where it can act to stabilise the cytoskeleton in advance of permanent repair by the production of new cytoskeletal proteins[5].

**Phosphorylation of FLNC near the interface modulates dissociation in concert with HSPB7**

Phosphorylation has been identified as a regulatory mechanism that influences the mobility of FLNC[14,15,48]. Notably, the phosphorylation of

serine and threonine residues on FLNC, for instance by protein kinase C, can have an influence along the length of the protein, rigidifying it and reducing its mobility[48]. This includes hindering the proteolytic cleavage between d23 and d24[14,48]; and inhibiting the binding with FILIP1 required to initiate degradation of FLNC[15]. We investigated the two sites of phosphorylation of FLNC associated with non-basal conditions: T2677 and Y2683. We chose them due to their proximity to the dimer interface within d24, reasoning that they might lead to a shift in the FLNC monomer-dimer equilibrium. Our results show phosphorylation at T2677 stabilises the homodimer, and phosphorylation at Y2683 weakens it. Remarkably, we observed the opposite effect on the FLNC:HSPB7 heterodimer. The combined effect of these two changes on the coupled equilibria is that, in the presence of HSPB7, stabilisation of the FLNC homodimer is increased in the case of pT2677, whereas its dissociation is heightened for pY2683 (Fig. 6C). The enhancement by HSPB7 of phosphorylation-based regulation adds strength to the hypothesis of interplay between molecular chaperones and kinase-mediated signalling pathways[49].

Phosphorylation at T2677 has been reported (and validated in our manual analysis of the data) in cardiovascular pathologies. Our insight predicts that this phosphorylation would act to stabilise the FLNC homodimer, reducing its mobility in the cell, and acting to solidify the cytoskeletal linkages that FLNC mediates. pY2683, instead, was identified in exercise-induced and cancer phospho-proteomics screens. In general, increased phosphorylation on tyrosine is common in cancer, due to the activation of tyrosine kinases, which leads to modulation of a variety of cellular activities[50,51]. Our data suggest that the weakened FLNC homodimer interface that we observed upon Y2683 phosphorylation may increase the mobility of FLNC, with potential pathological consequences. Interestingly, the genetic variant *FLNC* Y2683N (which is necessarily deficient in phosphorylation at this site) was reported

twice in the clinical genetic database ClinVar[52]; accession: VCV000850469.10. The first case was described non-specifically as 'disease of the cardiovascular system'; the second detailed skeletal and cardiac phenotypes ('Dilated Cardiomyopathy, Dominant Distal myopathy with posterior leg and anterior hand involvement, Hypertrophic cardiomyopathy 26, Myofibrillar myopathy 5'). In the absence of knowledge on biological function and lack of co-segregation data, the variant was classified as 'Variant of Unknown Significance' in ClinVar, but is an interesting target for studies in cellular or animal models.

While the focus of this current study is the structural and biophysical characterisation of FLNC homo- and hetero-dimerisation, our insights provide a mechanistic rationale for how this protein might switch between its multiple apparent functions, with the monomer representing a mobile state that is inherently incapable of crosslinking actin filamins within the cytoskeleton, but can translocate to sites where it is required, with the dimer being less mobile but able to provide cytoskeletal stabilisation through actin-crosslinking. This equilibrium therefore represents an opportunity to modulate the function of FLNC, with possible implications for both muscle disease and cancer. Future functional work will confirm the role of phosphorylation at T2677 and Y2683 for FLNC mobility and striated muscle physiology in animal and cellular models, the latter, e.g. based on induced pluripotent stem cell-derived cardiomyocytes[53]. In these approaches, the consequences of *FLNC* variants which exclusively carry the phosphorylation-deficient or -mimicking residues at each position will be studied. Similarly, our work motivates the identification and spatial-temporal regulation of protein kinases mediating the phosphorylation to understand the potential (patho-)physiological triggers of changes in FLNC mobility.

### HSPB7 evolved its unique interaction with FLNC through the loss of self-interaction

Our elucidation of the specific association made by HSPB7 with FLNC is unusual within the context of the current understanding of sHSPs. This degree of specificity and binding strength to a target protein has not been observed for a member of the sHSP family before, and it is also notable that the target is natively folded rather than partially unfolded or en route to aggregation[27,33,54]. Access to FLNC is facilitated by the ACD not binding other regions of the HSPB7 sequence, either inter- or intra-molecularly. As a result, HSPB7 is monomeric at low µM concentrations in vitro, comparable to the abundance at which it is found in cells that express it[55]. In line with this, even upon over-expression in HEK293 cells, HSPB7 was found to exist as a low molecular weight species[26].

Our phylogenetic analyses suggest this interaction is >400 million years old. The most likely evolutionary trajectory for HSPB7 is the duplication of an ancestral sHSP followed by degeneration of its oligomerisation interfaces. This degenerative loss then may have unmasked a latent ability to bind FLNC$_{d24}$, which is prevented in other sHSPs through their self-interactions, and consolidated by the presence of an additional hydrophobic residue (I102 in humans) that engages in the heterodimer interface. Given the promiscuous binding characteristics and broad tissue distribution of canonical sHSPs (e.g. HSPB1, HSPB5 in vertebrates), their duplication provides good opportunities for sub- and neo-functionalisation into chaperones tailored to the specific regulatory needs of a particular target protein. We therefore anticipate finding other such examples within the sHSP and wider molecular chaperone families.

## Methods
### Ethical statement
Experimental procedures were performed in accordance with the UK Home Office guidelines (project licences 30/2444 and 30/2977) and approved by the institutional review board (ACER, University of Oxford, UK).

### Protein expression and purification
Full-length human HSPB7 (UniProt ref: Q9UBY9-2, Isoform 2, A67 to AAHPTA, 175 residues) encoded in the pET23a vector was transformed into *E. coli* BL21(DE3) pLysS cells (Agilent) and expressed in LB media (with 0.5% glucose, 50 mg/mL ampicillin, and 37 mg/mL chloramphenicol) for 16 h at 22 °C following cell culture growth to OD600 between 0.6 and 0.8 and induction with 0.5 mM Isopropyl β-D-1-thiogalactopyranoside (IPTG). Cells were lysed using French Press Microfluidizer in 20 mM Tris, 1 mM EDTA, 5 mM β-mercaptoethanol (BME) at pH 7.4 with EDTA-free protease inhibitor cocktail. The lysates were centrifuged at 20,000 × *g* for 20 min at 4 °C. The resultant pellet was washed with the lysis buffer supplemented with 1% Triton X-100 twice, then once with 1 M urea, to remove membrane-associated contaminants. The protein was then solubilised in 6 M urea and 10 mM sodium phosphate at 4 °C, and refolded by removing urea stepwise in dialysis buffers containing 3 M, 2 M, and 1 M urea, with 100 mM NaCl, 20 mM Tris, and 5 mM BME (pH 7.5) buffering the process.

HSPB7$_{ACD}$ (residues 78–162 of isoform 2 UniProt ref: Q9UBY9-2, both WT and C131S forms), HSPB7$_{\Delta N}$ (residues 78–175 of isoform 2, WT and C131S) and FLNC$_{d24}$ (residues 2630–2724, WT and phosphomimic mutants T2677D and Y2683E) encoded in the pET28a(+) vector with an N-terminal His$_6$ tag were transformed into *E. coli* BL21(DE3) cells (Agilent). Cells were used to inoculate a 10 mL LB culture with kanamycin (30 µg/mL) at 37 °C overnight and transferred to a 1 L large culture with the same antibiotic. The culture was shaken at 37 °C until the OD600 reached 0.6–0.8, followed by induction with IPTG to a final concentration of 0.5 mM. After shaking overnight again, at 22 °C, the cells were harvested and lysed using a microfluidizer with the addition of EDTA-free protease inhibitor cocktail and buffer containing 300 mM NaCl, 50 mM Tris and 20 mM imidazole (pH 8.0). The lysates were then spun at 20,000 × *g* for 20 min at 4 °C and loaded onto a HisTrap HP column (Merck). The protein was eluted with buffer containing 300 mM NaCl, 50 mM Tris and 500 mM imidazole (pH 8.0) and dialysed to the loading buffer with TEV protease at room temperature overnight. Cleavage of the His$_6$ tag leaves a residual Gly-Ser overhang on the N-terminus of the constructs. The sample was again loaded onto a HisTrap HP column to remove the free His$_6$ tag and uncleaved proteins, and exchanged to PBS buffer with 5 mM Tris (2-carboxyethyl) phosphine (TCEP) for FLNC$_{d24}$ constructs or buffer containing 50 mM Tris (pH 8.0) for HSPB7 constructs. HSPB7 constructs were loaded onto a HiTrap Q anion exchange column (Merck) and eluted with a gradient of 150–200 mM NaCl in the elution buffer. The pure protein was then exchanged into PBS buffer with 5 mM TCEP for storage.

### Crystallisation and structure determination
Freshly prepared HSPB7$_{ACD}$$^{C131S}$ protein that had never been frozen was buffer-exchanged into 20 mM 4-(2-hydroxyethyl)-1-piperazineethanesulfonic acid (HEPES, pH 8.0), concentrated to 10 mg/mL and used for crystallisation. Crystallisation broad screens were prepared using an Art Robbins Phenix liquid handler and MRC SWISSCI 96-3 well sitting-drop vapour diffusion plates (Hampton Research, US). Drops containing 100:200 nL, 100:100 nL, and 200:100 nL (protein to well) were set with an 80 µL precipitant well volume. The crystal for structural determination of HSPB7$_{ACD}$ C131S was discovered and harvested after setting up for 10 days from a well containing 0.2 M magnesium formate dihydrate and 20% w/v polyethylene glycol (PEG) 3350 (JCSG-plus A5, Molecular Dimensions, protein: precipitant ratio 2:1). Crystals were cryoprotected using 10% glycerol in well solution, mounted on a nylon loop and then cryo-cooled by rapid plunging into liquid N$_2$. Diffraction data were collected and auto-processed at Diamond Light Source beamline I04. The diffraction data were integrated, merged and scaled automatically using Xia2 processing pipelines at Diamond Light Source[56]. The Phenix software package[57] was used for data analysis as follows. Molecular Replacement in PHASER using a search model generated with the Robetta comparative modelling

server[58], which was based on the HSPB6 (PDBID: 4JUS) structure, was used to obtain initial phases. Refinement was carried out using iterative cycles of fitting in COOT[59] and refinement sequentially in Phenix-refine, PDB-REDO[60] and CCP4 REFMAC[61]. The final structure has $R$ and $R_{free}$ values at 0.253 and 0.280, respectively.

For crystallisation of the HSPB7$_{ACD}$$^{C131S}$:FLNC$_{d24}$ heterodimer, both proteins (purified concurrently) were mixed at a ratio of 4:1 based on monomer and injected onto a gel-filtration column (Superdex 200, 16/600, Merck) equilibrated in 20 mM HEPES buffer (pH 8.0). The eluate corresponding to the heterodimer was concentrated to 10 mg/mL and used for crystallisation. Crystal screens were set as described above. Crystals appeared after one week in a condition containing 0.1 mM HEPES and 20% w/v PEG 8000, pH 7.0 (The Protein Complex Suite F1, Molecular Dimensions, protein: precipitant ratio 2:1). The crystal was harvested and flash-cooled without cryo-protectant. Diffraction data were collected at Diamond Light Source beamline I03, and auto-processed by Xia2 using the processing pipelines at Diamond Light Source. The FLNC$_{d24}$ chain in the heterodimer was phased using PHASER with FLNC$_{d24}$ (PDBID: 1V05) as the initial search model and was subsequently fed to CCP4 PHASER[61] with HSPB7$_{ACD}$$^{C131S}$ chain A to solve the heterodimer structure. Iterative cycles of fitting and refinement were carried out using COOT and Phenix-refine to $R$ and $R_{free}$ values of 0.224 and 0.291, respectively. Data collection and refinement statistics are shown in Supplementary Table 1.

## Quantitative native MS

All proteins except for the full-length HSPB7 were buffer-exchanged into 200 mM ammonium acetate (pH 6.9) using biospin (Bio-Rad); full-length HSPB7 was buffer-exchanged into 1 M ammonium acetate (pH 6.9). C-terminal peptides (Biomatik) were dissolved in 200 mM ammonium acetate (pH 6.9). Native MS was performed using nano-electrospray infusion in positive ion mode, using gold-coated capillaries that were prepared in-house, on a Synapt G1 (Micromass/Waters) modified for the transmission of intact non-covalent protein complexes. Experiments were performed according to standard native MS procedures[62], with key instrumental parameters as follows: capillary, 1.5 kV; sampling cone, 20–35 V; extraction cone, 3.0–3.3 V; backing pressure, 3.8–4.1 mbar; trap gas (argon) flow, 1.5–3.0 mL/min; trap, 6–8 V; transfer, 6–12 V. For the peptide titration experiments, acetonitrile was added to the source housing to help preserve non-covalent interaction[63].

Theoretical and measured masses are shown in Supplementary Table 2. Three technical repeats were collected for each ratio in the titration experiments, and peak intensities were extracted manually. We fitted the titration curves using either GraphPad Prism 8.0.1 or Python. The fitting of the ACD-peptide titrations used a Hill slope model for a 1:1 binding, with $B_{max}$ and $h$ values constrained to 1.0, while the competitive homo- and hetero-dimerisation was considered as a system of coupled equilibria (Supplementary Methods).

## Aggregation inhibition assay

Citrate synthase (CS) from the porcine heart (Sigma-Aldrich Co Ltd.) and HSPB7 constructs were buffer-exchanged into the reaction buffer with 40 mM HEPES and 1 mM DTT, pH 7.5. The final concentration of CS was maintained at 1 μM, while the concentrations of HSPB7 constructs were varied from 0.25 to 4 μM. The samples were transferred to 96-microwell plates, and aggregation of CS was induced at 45 °C and monitored through the change in apparent absorbance at 340 nm due to light scattering for at least two hours in a Fluostar Optima plate reader (BMG Lab Technologies). Three technical repeats were conducted for each ratio.

## Molecular Dynamics simulations

The crystal structures of wild-type FLNC$_{d24}$ homodimer (PDBID: 1V05) and FLNC$_{d24}$:HSPB7$_{ACD}$$^{C131S}$ heterodimer were used as the initial

structures for MD simulations. The structures of phosphorylated residues were modelled using Visual Molecular Dynamics (VMD). Each complex was immersed in a TIP3P periodic cubic water box extending 10 Å beyond the protein boundaries in every direction, neutralised with 0.15 M NaCl. Atomic interactions were described according to the Amber FF14SB force field[64], combined with the phosAA19SB force field[65] to describe phosphorylated residues (held at two negative unit charges).

MD simulations of all complexes were run using the NAMD2.14b2 engine[66]. All bonds were restrained using the SHAKE[67] algorithm, enabling a 2 fs timestep, and long-range electrostatic interactions were treated with Particle Mesh Ewald[68] with a 12 Å cutoff. The systems were first minimised via conjugate gradient, with 2000 steps for the wild-type dimers and 10,000 steps for modified dimers, then simulated in the nPT ensemble (300 K, 1 atm) for 0.5 ns with alpha carbons fixed by a 10 kcal mol$^{-1}$ restraint. Constant temperature and pressure were maintained by Langevin dynamics, with 1 ps$^{-1}$ as the damping constant, 200 fs as the piston period, and 50 fs as the piston decay. The pressure of each system was then equilibrated by removing the above restraints and running a 1 ns simulation in the nVT ensemble. Finally, for each system, a 1 μs (five 200 ns) production run in the nPT ensemble was carried out, with frames collected every 1 ns. All six systems remained stable during the production runs (Supplementary Fig. 7A).

Simulations were analysed with custom Python scripts using the MDAnalysis, numpy, and matplotlib packages[69,70]. For the hydrogen-bonding contact analyses, the cutoff values used for identifying hydrogen bonds were set as 1.2 Å for the donor-hydrogen distance, 3.0 Å for the hydrogen-acceptor distance and 150° as the donor-hydrogen-acceptor angle. The hydrogen bond occupancy of any given pair of residues was calculated as a percentage, taking into account the possibility that residue pairs could form more than one hydrogen bond.

To identify the contact surface within FLNC$_{d24}$ homo- and heterodimers, we calculated both the average shortest distance and minimum shortest distance from each residue in FLNC$_{d24}$ to the partner monomer during each frame of the MD trajectories. The shortest distance is defined as being between any heavy atom on the FLNC$_{d24}$ residue in question and any heavy atom on the partner monomer. To display this on a surface rendering of FLNC$_{d24}$, we coloured each residue with the saturation negatively linearly correlated to the value of the average shortest distance from the partner monomer.

## HDX-MS

HSPB7$_{ACD}$, FLNC$_{d24}$ and the heterodimer were all prepared in 40 mM HEPES at pH 7.5. To form the heterodimer, HSPB7$_{ACD}$ and FLNC$_{d24}$ were mixed at 1:1 (25 μM: 25 μM, for HSPB7$_{ACD}$ measurement) or 4:1 (50 μM: 12.5 μM, for FLNC$_{d24}$ measurement) molar ratios. To carry out the HDX-MS experiments, 5 μL sample volumes were each exposed to labelling in 55 μL of D$_2$O buffer (40 mM HEPES, pD 7.5) for a time course of 30, 60, 500, 1000 and 5000 s, followed by mixing with one equivalent of pre-chilled quench solution (60 μL, 40 mM HEPES, 5 mM TCEP, pD 2.0). 80 μL of each of the quenched samples was then injected in turn into a nanoACQUITY UPLC System (Waters) and digested online using an Enzymate™ BEH Pepsin Column (2.1 × 30 mm, Waters) at room temperature. The digested peptides were trapped by a BEH C18 trap column (1.7 μm, 2.1 × 5 mm, Waters) for 3 min for desalting. A BEH C18 analytical column (1.7 μm, 1 × 100 mm, Waters) was then used to separate the peptides using a linear gradient from 3% to 35% of acetonitrile with 0.1 % formic acid at a flow rate of 40 μL/min. To minimise back-exchange, the temperature was kept below 4 °C after digestion. A wash buffer of 1.5 M guanidinium chloride, pH 2.8, 4% acetonitrile, 0.8% formic acid was used to clean the pepsin column between each run. MS data were obtained on a Synapt G2-Si (Micromass/Waters) operating in positive ion mode, and fragmentation

spectra were obtained in MS$^E$ mode. Peptides from undeuterated samples were analysed and identified using ProteinLynx Global Server 2.5.1 (Waters). DynamX 3.0 (Waters) and Deuteros 2.0[71] were used to analyse and visualise the HDX-MS data. The peptide-level significance test (which is two-tailed) was used as the model, with the alpha value set at 0.01 (i.e. 99% confidence).

## Phylogenetic analyses and ancestral sequence reconstruction

To infer the phylogenetic tree of HSPB7 and homologues, amino-acid sequences were gathered by online BLASTP on 19 September 2023 using human (NP_055239.1) and elephant shark (XP_007905160.1) HSPB7 amino-acid sequences as initial queries. Gathered HSPB7 sequences were further used to interrogate genomes from representative taxa in each major clade, resulting in an expanded set of 471 HSPB amino-acid sequences spanning Bilateria and Cnidaria. To distinguish between homologues, all HSPB ACD domains were first aligned using MUSCLE (v.3.8.31) before tree inference by FastTree (v.2.1.11). Sequences clustering together in each HSPB clade were manually curated to remove lineage-specific indels before profile-alignment with MAFFT (v7.505).

The maximum likelihood (ML) phylogeny of all HSPB ACD domains was inferred using RaxML-NG (v.1.1) and the Jones-Taylor-Thornton model, as determined by automatic best-fit evolutionary model selection in IQ-TREE (v.2.2.0.3), with gamma-distributed among-site rate variation and fixed base frequencies. Lastly, to obtain the final tree, hagfish and *Styela* HSPB7 sequences were constrained to the base of vertebrates due to the incongruence of their positions in the ML tree with known species relationships.

Ancestral sequences and posterior probabilities (PP) of ancestral states were reconstructed at internal nodes leading to the HSPB7 lineage in our constrained tree as implemented in IQ-TREE, using the same evolutionary model as per tree inference. Gap assignment of ancestral sequences was determined using Fitch parsimony with PastML (v.1.9.34) given our final tree. Ancestral sequences contain states with the highest PP at all sites selected. For alternative ancestors, states with the second best PP if PP > 0.20 at each site were selected, with all other sites containing ML states.

The extant FLNC domain 24 sequences were gathered using the online BLASTP tool on 4 January 2024. The sequence for human FLNC d24 (NP_055239.1) was used as a query sequence, and the search was restricted to the organisms selected as representative of their taxa. From the sequences found, the hit with the highest percentage identity was chosen as the FLNC d24 homologue of that organism. The resulting sequence IDs are: elephant shark (XP_042191161.1), *Styela* (XP_039265544.1), zebrafish (AAH90688.1), *Xenopus* (AAH99062.1), green anole (XP_016864456.1), chicken (NP_989904.1).

## In silico competitive-binding assay

All structure predictions were generated using ColabFold[72], a fast and reliable software that combines the fast homology search of MMseqs2 with AlphaFold2[73]. We ran AlphaFold locally on a Dell Precision 5820 workstation installed with an Nvidia RTX4000 GPU card. We used 10 recycles through the network to improve the prediction and the *--amber* flag to relax the predicted structure using the AMBER force field. The structures generated were ranked using pTM (estimate of the Template Modelling score). To generate multiple structures, we used different seeds, and the best-ranked structure was used for visualisation and further investigation.

Because AlphaFold inevitably produces a structure of a protein complex when two (or more) chains are inputted, distinguishing false from true positives is a major challenge. Hence, we designed a competition assay where we provided two chains of each of HSPB7 and FLNC as input to AlphaFold. Therefore, homo-dimerisation (giving HSPB7$_2$ and FLNC$_2$) and hetero-dimerisation (giving 2 copies of FLNC:HSPB7) outcomes are both possible. The resulting models from the competition were inspected and assigned as homo- or heterodimers.

## Mouse models of biomechanical stress

Experimental procedures were performed in accordance with the UK Home Office guidelines (project licences 30/2444 and 30/2977) and approved by the institutional review board (ACER, University of Oxford, UK).

Mice (*Mus musculus*, on C57BL/6J background) were housed in specific pathogen–free conditions, with the only reported positives on health screening over the entire time course of these studies being for *Tritrichomonas* sp. and *Entamoeba* spp. All animals were housed at 19–21 °C, at 55 ± 10% humidity, in social groups (unless they had surgical interventions, then they were singly housed after the intervention). They were provided with food (Irradiated Global 16% rodent diet T26.16MI, Envigo RMS, UK, Ltd) and water *ad libitum*, and maintained on a 12-h light/12-h dark cycle (150–200 lux cool white light-emitting diode light, measured at the cage floor).

MLP KO mice[74] were backcrossed onto a C57BL/6J background (purchased from Harlan, UK, Ltd) for more than six generations before generating homozygous MLP KO mice. Genotyping for the knockout allele was performed as above using the following primer pair: 5′-CCTTCTATCGCCTTCTTGACGAG-3′ and 5′- CTCATACTCGGAACTTGG G-3′, the wild-type allele was probed for with the following primer pair: 5′-CAGGCTGTCCCCTAGACCTC-3′ and 5′ −GAACCACCAACAGACAGTA GTAGG-3′. All genotyping was performed from ear biopsies with REDExtract-N-Amp Tissue PCR Kit (Sigma). Age- and sex-matched wild-type C57BL/6J were obtained from Harlan.

For chronic adrenergic stimulation by IsoPE, ~3-month-old animals (bred on C57BL/6JOlaHsd background in-house, Envigo RMS, UK, Ltd) received an injection of buprenorphine (0.05 mg/kg body weight of 0.3 mg/mL Vetergesic Ceva) subcutaneously for pain prevention before the surgical procedure. Osmotic minipumps (ALZET 1002) for drug administration were implanted subcutaneously in a sterile mid-scapular incision procedure under general anaesthesia (constant supply at 2% v/v in 1 L/min oxygen). Treatment groups received IsoPE (in hydrochloride forms, Sigma-Aldrich) at a concentration of 15 mg kg$^{-1}$ body weight each in 0.9% NaCl (Vetivex 1, Dechra) at a constant flow rate of 0.25 µl/h per day for 14 days in total. Control group ('saline') received 0.9% NaCl as a placebo via osmotic minipumps at the same flow rate and for the same amount of time. All animals were monitored daily according to the PPL guidelines. Animals were sacrificed and organs harvested on day 14.

For TAC, ~2-month-old mice (bred onto C57BL/6JOlaHsd in-house, Envigo RMS, UK, Ltd; body weight 24 ± 3 g) were anaesthetised with isoflurane (constant supply at 2% v/v in 1 L/min oxygen), intubated, and a trans-sternal thoracotomy performed. The transverse aorta was constricted with a 7-0 polypropylene monofilament suture (Ethicon) tied against a 27-gauge needle. In sham-operated mice, the aortic arch was dissected, but no suture was tied. Mice were given subcutaneous buprenorphine (0.8 mg/kg) for pain relief. Mice were sacrificed and hearts harvested 14 days after intervention.

Only male mice were used to minimise variability for treatment experiments. For treatment experiments, a randomised block design was used. Operators were blinded during processing and analysis of mouse samples, and no animals were excluded from the study. For organ harvest, animals were culled by a schedule 1 method (i.e. cervical dislocation followed by confirmation of death via cessation of circulation); no drugs or chemicals were used for euthanasia. Hearts were dissected using sterile surgical tools. Whole hearts were washed in PBS. Ventricular tissue was snap frozen in liquid nitrogen and stored at −80 °C for future analysis.

Mouse hearts used here were generated in previous studies[75,76], and two hearts from each group were selected randomly. Age and sex for each mouse are given in Supplementary Table 4.

## Western blotting, co-immunoprecipitation and immunofluorescence

Western blotting (WB), co-immunoprecipitations and immuno-fluorescence (IF) on cryosections from mouse cardiac tissue were essentially performed as described previously[16].

For Western blotting, tissue powder (BioPulverizer 0.1–1 g) was incubated twice with SDS-sample buffer (62.5 mM Tris-Cl, pH 6.8, 2% sodium dodecyl sulfate, 0.02% bromophenol blue, 100 mM dithiothreitol, 6% glycerol), heated at 65 °C for 15 min and sonicated. Samples were diluted in 1.5× SDS-sample buffer. Gel electrophoresis was performed using Mini-PROTEAN TGX Precast Gels (4–15%, Bio-Rad), which were transferred onto nitrocellulose membranes using the iBlot2 system (ThermoFisher Scientific). Membranes were blocked with 5% milk powder in TBS + 0.1% Tween 20 (TBS-T) and incubated with primary antibodies at 4 °C overnight (see below). Secondary antibody (donkey anti-rabbit horseradish peroxidase conjugated, GE Healthcare) was added (1:10,000 in 1% milk-TBS-T) and left for a minimum of one hour. Membranes were rinsed in TBS-T before three 20-min washes in TBS (0.5% Tween 20). Membranes were incubated with SuperSignal West Pico PLUS Chemiluminescent Substrate or SuperSignal West Dura Extended Duration Substrate (ThermoFisher Scientific) and signals visualised on a ChemiDoc MP system (Bio-Rad).

The following antibodies were used: HSPB7 (raised in rabbit), 15700-1-AP (Proteintech, Lot 00006996), for WB at 1:1000 v/v dilution; HSPB7 (raised in rabbit), NBP1-84334 (Novus, Lot 000012270), for IF at 1:50 v/v dilution; FLNC (raised in mouse, IgA, obtained in 2018), RR90[4], for IF at 1:50 v/v dilution; FLNC (raised in sheep), 1899 (Medical Research Council Protein Phosphorylation and Ubiquitinylation Unit, Reagents and Services, University of Dundee, purchased in 2017, 0.26 mg/mL) for WB at 1:2000 v/v dilution; GAPDH (raised in rabbit), ABS16 (Merck, Lot 2745933), for WB at 1:3000 v/v dilution.

For co-immunoprecipitation, tissue lysates (wild-type and MLP KO hearts) were prepared using IP buffer (1% Triton X-100, 0.5% NP-40, 20 mM Tris-HCl (pH 7.6), 138 mM sodium chloride, 5 mM dithiothreitol, 100 mM potassium chloride, 60 mM octyl-β-glucopyranoside, phosphatase and protease inhibitors, Roche), and incubated on ice for 60 min. After spinning and pre-clearing with Protein G beads (Sigma, 1 h at 4 °C), lysates (900 μg total protein) were precipitated with 2 μg anti-FLNC antibody 1899 in 500 μL buffer overnight on ice. As a control, 2 μg isotype control antibody was used (Sigma, sheep IgG, purchased March 2019). 25 μL Protein G beads (Sigma) were added, incubated for 1 h 15 min at 4 °C (shaking). Beads were washed with IP buffer and bound proteins eluted twice with 25 μL of SDS-sample buffer and subsequent heating (100 °C, 3 min). Western blots were performed as described above, using VeriBlot (Abcam, 1:1000, Lot 1035165-12) as secondary antibody for HSPB7 detection.

For immunofluorescence on cryosections, sectioning was done on OCT (VWR) embedded tissue on a Cryotome FSE (Thermo Scientific; 10 μm thickness). Staining and microscopy of sections were performed as described[77], using primary antibodies given above and goat anti-mouse IgA Heavy Chain Antibody DyLight 488 conjugated (A90-103F, Bethyl, 1:25) and goat anti-rabbit (Fab)2 fragment AlexaFluor 568 conjugated (A-11019, Molecular Probes, 1:100) secondary antibodies. Mounted slides were imaged on a Leica SP5 microscope, equipped with Argon and HeliumNeon lasers and a 63× NA1.4 oil objective.

### Reporting summary

Further information on research design is available in the Nature Portfolio Reporting Summary linked to this article.

## Data availability

The data on which this manuscript is based is made freely available through the ORA-Data repository [https://doi.org/10.5287/ora-qmrz8mxnz]. The structural files have been deposited in the Protein Data Bank (PDB) under accession codes: 8PAO (FLNC_d24:HspB7_ACD^C131S),

8RHA (HSPB7_ACD^C131S). Data underlying the plots in Fig. 3E, Fig. 4B, Fig. 6A, and Supplementary Fig. 1, Supplementary Fig. 2D, Supplementary Fig. 3B, and Supplementary Fig. 6B are given in the Source Data spreadsheet. Supplementary Data 1 contains all the hydrogen bond contact data from MD simulations. Other PDB files associated with this study are: 1V05, 4MJH. Source data are provided with this paper.

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

## Acknowledgements

This work was supported by the Medical Research Council [MR/V009540/1 to Z.W., S.B.S., K.G. and J.L.P.B.]; British Heart Foundation (BHF) [FS/12/40/29712 to K.G., IA/F/23/275037 to S.B.S. and K.G.]; the Oxford BHF Centre of Research Excellence [RE/13/1/30181 to K.G.]; the Wellcome Trust [201543/B/16/Z to K.G.]; the Leverhulme trust [grant no. RPG-2021-246 to D.S., J.Z.Y.N., G.K.A.H. and J.L.P.B.], the German Research Foundation (DFG) [FOR 2743, project P09 (WA1598/6) to B.W.] and [FOR 2743 - FU339/11-2 and FU339/13–1 to D.O.F.]. The Institute of Cardiovascular Sciences, University of Birmingham, has received an Accelerator Award by the British Heart Foundation [AA/18/2/34218]. J.R.B. acknowledges support from the Royal Society University Research Fellowship. We also thank the staff of I03 and I04 at Diamond Light Source for their support.

## Author contributions

Conceptualisation: Z.W., M.P.C., K.D.-C., D.O.F., B.W., G.K.A.H., K.G. and J.L.P.B. Investigation: Z.W., G.C., M.P.C., X.Q., S.B.S., J.Z.Y.N., N.S., A.J.A., C.H., J.Z., J.B. and T.R.A. Formal analysis: Z.W., G.C., X.Q., D.S., J.Z.Y.N., M.M., P.R., H.S., J.R.B., M.T.D. and T.M.A. Writing – original draft: Z.W., K.G. and J.L.P.B. Writing – review & editing: all authors. Supervision: C.J.S., B.W., M.T.D., G.K.A.H., C.V.R., K.G. and J.L.P.B.

## Competing interests

The authors declare no competing interests

## Additional information

¹Department of Chemistry, Dorothy Crowfoot Hodgkin Building, University of Oxford, Oxford, UK. ²Kavli Institute for Nanoscience Discovery, University of Oxford, Oxford, UK. ³School of Molecular and Cellular Biology, Faculty of Biological Sciences, University of Leeds, Leeds, UK. ⁴Cardiovascular Sciences, School of Medical Sciences, University of Birmingham, Birmingham, UK. ⁵Evolutionary Biochemistry Group, Max Planck Institute for Terrestrial Microbiology, Marburg, Germany. ⁶Center of Biological Design, Berlin Institute of Health at Charité, Universitätsmedizin Berlin, Berlin, Germany. ⁷Division of Cardiovascular Medicine, Radcliffe Department of Medicine and British Heart Foundation Centre of Research Excellence Oxford, University of Oxford, Oxford, UK. ⁸Biochemistry II, Theodor Boveri-Institute, Biocenter, Chemistry and Pharmacy, University of Würzburg, Würzburg, Germany. ⁹Department of Chemistry,

Chemistry Research Laboratory, Oxford, UK. [10]Enzymology and Applied Biocatalysis Research Center, Faculty of Chemistry and Chemical Engineering, Babes-Bolyai University, Cluj-Napoca, Romania. [11]Diamond Light Source, Harwell Science and Innovation Campus, Oxfordshire, UK. [12]Helmholtz Munich, Molecular Targets and Therapeutics Center, Institute of Structural Biology, Neuherberg, Germany. [13]Technical University of Munich, TUM School of Natural Sciences, Department of Bioscience, Bavarian NMR Center, Garching, Germany. [14]Ineos Oxford Institute for Antimicrobial Research, University of Oxford, Oxford, UK. [15]Department of Biology, University of Oxford, Oxford, UK. [16]European Molecular Biology Laboratory, Grenoble, France. [17]Department of Structural and Computational Biology, Max Perutz Labs, University of Vienna, Vienna, Austria. [18]Institute for Cell Biology, University of Bonn, Bonn, Germany. [19]Department of Physics, Durham University, Durham, UK. [20]School of Informatics and EaStCHEM School of Chemistry, University of Edinburgh, Edinburgh, UK. [21]Biomolecular Interaction Centre and School of Physical and Chemical Sciences, University of Canterbury, Christchurch, New Zealand. [22]Present address: Department of Chemistry, Philipps-University Marburg, Marburg, Germany. [23]Present address: Center for Synthetic Microbiology, Philipps-University Marburg, Marburg, Germany. [24]These authors contributed equally: Guodong Cao, Miranda P. Collier, Xingyu Qiu.
✉e-mail: k.gehmlich@bham.ac.uk; justin.benesch@chem.ox.ac.uk

