## [Transparent Peer Review file · Nature Communications]

Filamin C dimerisation is regulated by HSPB7

Corresponding Author: Professor Justin Benesch

Version 0:

Reviewer comments:

Reviewer #1

(Remarks to the Author)

The authors looked at possible interactions of HSPB7, which is a molecular chaperone in myocardium, with filamin C, an actin cross-linking protein that plays a role in cardiac contractility and in responses to certain biomechanical stresses that arise in heart diseases, including pressure overload.

The authors found that filamin C and HSPB7 spatially associate in cardiac tissues during certain types of experimentally induced stress. The pathophysiological significance of this finding is not well demonstrated as the authors do not provide high resolution imaging (e.g., by proximity ligation analysis) or quantification of their two colour immunostaining to validate the spatial co-localization of their main proteins of interest or how the co-localization may change as a result of pressure overload or disease.

The authors went on to show that filamin C and HSPB7 form a heterodimer, which competes out homodimers formed by filamin C. The authors then speculate that the hetero-dimer could prevent filamin C homodimers to cross-link actin filaments in the contractile unit and altering its mobility (by diffusion). None of these speculations have been subject to experimental enquiry. The demonstration of these alterations of actin cross-linking and the structural alterations that would occur in the cardiac myofilament unit are needed to show that the interesting structural findings reported here actually have physiological relevance.

Of fundamental interest in the context of protein structure, the authors found that phosphorylation of filamin C (at T2677) enhances the formation of homodimers and is also coincident with the dimer interface. In contrast, phosphorylation of Y2683, which is also located near the dimer interface, exerts an opposing effect to promote the formation of heterodimers (i.e., filamin C-HSPB7). This part of the data set is well-performed but without stronger physiological significance (e.g., what is the effect of induced mutations in mouse of T2677 and Y2683 or spontaneous mutations in human on cardiac function; what structural and localization data can be provided from cardiac tissues in vivo), the physiological significance of these findings is not obvious.

The authors then conducted evolutionary analysis and ancestral sequence reconstruction that suggest the protein-protein interaction that they have discovered may have evolved at around the same time as the era of primitive heart evolution in chordates.

While the research is of considerable interest to our general knowledge of protein structure/function and to the very interesting proteins that have been examined here, the research presented does not provide sufficient connection to cardiac physiology and how filamin C alters its stabilizing roles in cardiomyocytes to contribute to normal and pathological contractile function and structure.

Reviewer #2

(Remarks to the Author)

The manuscript deals with interactions between FLNC and HSPB7 (cardiac-specific molecular chaperone). The authors report the formation of a heterodimer of FLNC and HSPB7 under biomechanical stress and have solved the X-Ray structure for the same. More detailed analysis compares the FLNC homo-dimer with the FLNC-HSPB7 heterodimer. Notably, the

dimer interface is practically the same, which brings in the question of competitive binding and regulation. Phosphorylation of two specific residues, T2677 and Y2683, both at the dimer interface is examined, and found to have opposite effects. Molecular dynamics simulations were carried out to explain the phosphorylation dependent modulation of the interactions across the homodimer and heterodimer interface. The results show that the WT homodimer is stabilized by inter-chain hydrogen bonds between E2681 and Y2683. This is lost in the pY2683 variant. In contrast, the stabilizing effect of pT2677 was traced to a salt-bridge with R2690 across the dimer interface only in the phosphorylated form. The interactions are altered in case of heterodimers. Hydrogen bonds between G2674 and N107 across the heterodimer interface is lost upon phosphorylation of T2677. This is attributed to a twist in the loop containing pT2677. In case of the PY2683 heterodimer, multiple stabilizing interactions across the dimer interface are noticed in the phosphorylated form (including the phosphorylated Y2683 residue itself).

Overall the results are noteworthy and the methodology is sound. Here are the specific comments/suggestions regarding the molecular dynamics part of the study.

- (i) Specify the charges on the phosphorylated residues and provide the reference for the Forcefields.
- (ii) Hydrogen bond occupancy should be directly stated (not just pictorially, but in numbers) in the important cases.
- (iii) In the comparison between WT and pT2677 heterodimer, the authors state that the loss of the hydrogen bond (N107-G2674) is due to a twist in the loop in which it (pT2677) resides. However the picture (Figure 5C) does not quite explain it. Rather it shows N107 to have moved away. Is it possible that pT2674 forms other interactions nearby that yanks away the loop? MD trajectories can clarify.
- (iv) Figure 6 is not explained adequately. For example, does it only show the WT? Are there corresponding plots for the variants or were they ignored because the interface distances were overlapping? In some cases, there were visible differences between the homodimer and heterodimer. So what are these residues, do they correspond to the residues involved in the hydrogen bonds shown in Figure 5?
- (v) References for the SHAKE algorithm and Particle Mesh Ewald method are missing.
- (vi) I am assuming that all the six systems simulated remained stable. There should be a statement about basic stability of the dimers.

Reviewer #3

(Remarks to the Author)

The manuscript by Wang et al, "Cardiac stress leads to regulation of Filamin C dimerisation via 1 an ancient phosphorylation-modulated interaction with HSPB7" examines the interaction between FLNC and HSPB7 on a molecular level, as well as its regulation through phosphorylation.

As noted by the authors HSPB7 is required for cardiac development and loss of HSPB7 leads to aggregation of FLNC and is embryonically lethal due to cardiac defects.

The authors propose that HSPB7 is specialised in its function on the interaction with FLNC and hence explore this interaction in detail using cell biological, biochemical and structural biology experiments. In mouse models they find that biomechanical stress leads to upregulation of the interaction. They go on to characterize further the structure/function relationship of HSPB7. They identify that HSPB7 interaction outcompetes the homo-dimerization of FLNC and that this is modulated through phosphorylation at either T2677 and Y2683 which either strengthen or weaken the homo-dimerization, respectively.

My only comment relates to the choice of the phosphorylation sites. I do not doubt the data to show changes in homo-dimerization, and the selection of the sites, based on the proximity to the dimerization site is logical. The evidence for the phosphorylation comes from manual re-analysis of deposited data sets, which are however from skeletal muscle and related to SARS-CoV-2 Infection and as such there is a lack of data about this phosphorylation in the heart. In this sense, it would be great if the authors could discuss if 1) these are the only potential phosphorylation sites in this region; 2) if the sequence reveals anything about potential kinases responsible for the phosphorylation and 3) if there are any cardiac phospho-proteomic data sets that show either of these phosphorylation sites.

Additionally there are a few typos (e.g. in the abstract 'important of physiological', or 'whose variants are causative')

Apart from this, the study is expertly executed, combining interdisciplinary data to elucidate the interaction between HSPB7 and FLNC and the mechanism of this atypical chaperone, which will be interesting to a wider audience and is overall a great fit for Nature Communications.

Reviewer #4

(Remarks to the Author)

This is a lovely body of work incorporating some really nice structural analysis of FLNC and its binding partners, looking at the potential structural effects of phosphorylation at two specific residues. However, I do have some concerns that the authors are currently over-selling the importance of this story from a biological perspective in light of the data that they present. They are quite possibly correct in their conclusions, but they do not provide the definitive evidence.

The manuscript title states that 'cardiac stress leads to ...via and ancient phosphorylation-modulated interaction...' – while the authors have shown how phosphorylation likely changes dimerization (based on phosphomimetics and MD simulations, see later points), and that complex formation changes in their mouse models of stress, they do not actually appear to have shown a direct link, in that they have not demonstrated in their cardiac model that these sites are phosphorylated, and

differences in the phosphorylation states of these sites in the different complexes. In the absence of this specific piece of data (which could be obtained using either antibody or MS-based approaches), I would ask that the authors tone down the definitive statements in the title and the abstract.

Please provide the evidence that pT2677 is associated with cardiac stress. I note that it was identified in previous studies (refs. 39/40) but it would be useful to include the evidence of its upregulation under these conditions to support the bold statements being made in the manuscript title and abstract, and link the structural work presented here to the underlying biology. Likewise with pY2683. If the authors are able to demonstrate phosphorylation of these two sites in their own systems, then this becomes obsolete.

P13 - Given that the authors are re-analysing published datasets, it would be useful to provide the data deposition (PDB?_ numbers) for reference.

While D/E are often used as phosphomimetics, it is possible to see different effects depending on which of these is used. Given their interpretation of the differential effects of phosphorylation of these two residues, it is therefore important to check that this is not purely the result of the different amino acid substitutions that have been used for T2677 (D) and Y2683 (E). A better experiment would be to do site specific phosphate incorporation using codon expansion, but these are potentially complicated experiments to set up and in their absence, complementary use of the different (pseudo)mimicking acidic residue is important to avoid potential mis-interpretation of the findings (although their conclusion appear to be supported by the MD simulations).

Can the authors give any suggestions as to what enzymes may be regulating these phosphorylation events which appear to play such crucial structural roles?

Minor comments - Abstract needs to be read through carefully – there are some minor grammatical errors.

'variety of important'

'Variants are causative'

Version 1:

Reviewer comments:

Reviewer #1

(Remarks to the Author)

The authors have provided one of the most conscientious, thorough, insightful and straightforward set of responses to the questions raised by the four reviewers. This level of presentation and clarity of purpose is really terrific and the paper is, in my estimation, really improved by the authors substantial efforts. Their data have advanced the filamin (C) field and their work in relating this to cardiomyopathy and cardiac function are much-appreciated.

Reviewer #2

(Remarks to the Author)

I am satisfied with the clarifications provided by the authors regarding the molecular simulations.
I have no other questions.

Reviewer #3

(Remarks to the Author)

The authors have made edits to the document to address the reviewers comments and I'm happy to support publication.

Reviewer #4

(Remarks to the Author)

The authors have addressed my comments to the best of their ability, providing much additional supporting data and toning down some of the statements in-line with the evidence provided.

Response to Reviewers

We are very grateful to the reviewers for their thoughtful consideration of our manuscript *NCOMMS-24-02273-T*, and their constructive feedback. We are delighted by their overall positive assessment of structure-function investigation into the human cardiac specific molecular chaperone HSPB7 and cytoskeletal protein FLNC. We recognize that the reviewers raise valid critiques and welcome the opportunity to revise the manuscript accordingly. Below, we have responded to each point made by the reviewers and indicated all additions and adjustments to the manuscript. Bold text in brackets [] refers to new phrasing or information, and parentheses () refer to material also in the original submission. We have also included a form of the revised manuscript in which all changes and additions are highlighted.

Reviewer 1

1.1 The pathophysiological significance of this finding is not well demonstrated as the authors do not provide high resolution imaging (e.g., by proximity ligation analysis) or quantification of their two colour immunostaining to validate the spatial co-localization of their main proteins of interest or how the co-localization may change as a result of pressure overload or disease.

We have taken up the excellent suggestion of the reviewer and quantified the immunostaining results. The analysis is included in the Supplementary Information [**new Supplementary Fig. 1, and Supplementary Methods**]. The Pearson's coefficients for co-localisation of filamin C and HSPB7 are significantly higher in two of our biomechanical stress models, MLP KO and TAC, compared to their respective controls (WT and Sham). For the IsoPE model of stress there is an increased Pearson's coefficient already in the Saline controls and hence no change compared to the IsoPE group. This fits with the qualitative observations in the immuno-fluorescence data, namely increased signal of HSPB7 at intercalated discs of the Saline group, compared to WT and Sham control groups (**Fig. 1C**), and is most likely caused by the surgical implantation of a Saline osmotic minipump. We have included these analyses [**Results, p6-7**].

1.2 The authors then speculate that the hetero-dimer could prevent filamin C homodimers to cross-link actin filaments in the contractile unit and altering its mobility (by diffusion). None of these speculations have been subject to experimental enquiry. The demonstration of these alterations of actin cross-linking and the structural alterations that would occur in the cardiac myofilament unit are needed to show that the interesting structural findings reported here actually have physiological relevance.

We agree with the reviewer that molecular mobility experiments are an attractive means to confirm that that binding by FLNC of HSPB7 changes its dynamic behaviour; the hetero-dimer (FLNC-HSPB7) is expected to be more mobile than the FLNC homo-dimer. Fluorescence recovery after photo-bleaching (FRAP) experiments can be used to assess the mobility of GFP-tagged FLNC, and we have performed such experiments successfully in the past¹. However, the obvious experiment – assessing FLNC mobility in HSPB7 knock-out cells by FRAP – is confounded by two issues: the global HSPB7 knockout is embryonic lethal at E12.5² and the cardiac-specific HSPB7 knockout leads to the formation of FLNC aggregates³, which confound assessment of FLNC dynamics. Hence, neither model is suitable for FRAP experiments.

1.3 This part of the data set is well-performed but without stronger physiological significance (e.g., what is the effect of induced mutations in mouse of T2677 and Y2683 or spontaneous mutations in human on cardiac function; what structural and localization data can be provided from cardiac tissues *in vivo*), the physiological significance of these findings is not obvious.

The reviewer asks what the *in vivo* consequences of phosphorylation-deficient or mimicking mutations at either site are. In terms of spontaneous human variations at the residues, according to GnomAD there is a once-reported phospho-deficient mutation at residue T2677 (T2677N, highlighted in red):

Variant ID	Source	HGVS Consequence	VEP Annotation	LoF Curation	Clinical Significance	Flags	Allele Count	Allele Number	Allele Frequency	Number of Homozygotes
7-128858359-G-A		p.Val2672Met	missense		Uncertain significance		6	1565474	3.83e-6	0
7-128858364-C-G		p.His2673Gln	missense		Uncertain significance		33	1571912	2.10e-5	0
7-128858365-G-A		p.Gly2674Ser	missense		Uncertain significance		36	1573522	2.29e-5	0
7-128858368-C-T		p.Pro2675Ser	missense		Uncertain significance		4	1581570	2.53e-6	0
7-128858371-A-G		p.Lys2676Glu	missense		Uncertain significance		3	1579256	1.90e-6	0
7-128858372-A-G		p.Lys2676Arg	missense				2	1585804	1.26e-6	0
7-128858372-G-C		p.Lys2676Asn	missense				1	1574772	6.35e-7	0
7-128858375-C-A		p.Thr2677Asn	missense				1	1583804	6.31e-7	0
7-128858375-C-G		p.Thr2677Ser	missense				1	1583806	6.31e-7	0

https://gnomad.broadinstitute.org/gene/ENSG00000128591?dataset=gnomad_r4 (accessed 12/7/24)

Given the heterozygous status, lack of data on this individual, late onset and incomplete penetrance of cardiac diseases, we cannot conclude anything from this occurrence.

For Y2683, nine cases with a heterozygous variant *FLNC* Y2683N have been reported (highlighted in red):

Variant ID	Source	HGVS Consequence	VEP Annotation	LoF Curation	Clinical Significance	Flags	Allele Count	Allele Number	Allele Frequency	Number of Homozygotes
7-128858371-A-G		p.Lys2676Glu	missense		Uncertain significance		3	1579256	1.90e-6	0
7-128858372-A-G		p.Lys2676Arg	missense				2	1585804	1.26e-6	0
7-128858373-G-C		p.Lys2676Asn	missense				1	1574772	6.35e-7	0
7-128858375-C-A		p.Thr2677Asn	missense				1	1583804	6.31e-7	0
7-128858375-C-G		p.Thr2677Ser	missense				1	1583806	6.31e-7	0
7-128858392-T-A		p.Tyr2683Asn	missense		Uncertain significance		9	1594542	5.64e-6	0
7-128858395-G-A		p.Val2684Met	missense		Uncertain significance		10	1596244	6.26e-6	0
7-128858404-A-G		p.Met2687Val	missense		Uncertain significance		9	1603686	5.61e-6	0

https://gnomad.broadinstitute.org/gene/ENSG00000128591?dataset=gnomad_r4 (accessed 12/7/24)

The minor allelic frequency 5.64×10^{-6} would be in keeping with a rare pathogenic role in disease (anything higher than the threshold of 10^{-4} would be considered too frequent to cause cardiomyopathies or myopathies). Of note, this variant has a ClinVar annotation, with two separate cases reported (Accession: VCV000850469.10, <https://www.ncbi.nlm.nih.gov/clinvar/variation/850469/>). Clinical description is vague for one ('Cardiovascular system'), but details skeletal and cardiac phenotypes in the other ('Dilated Cardiomyopathy, Dominant Distal myopathy with posterior leg and anterior hand involvement, Hypertrophic cardiomyopathy 26, Myofibrillar myopathy 5'). Based on the lack of knowledge on biological function and lack of co-segregation data, the variant is currently classified as a 'Variant of Unknown Significance' in ClinVar. However, it could be causative for the phenotypes observed. We have added a comment on this in the discussion of the manuscript [Discussion, p21].

We agree with the reviewer that characterization of (in particular) *FLNC* Y2783N would be of extreme interest and might even aid the diagnosis and management of patients. Such a project could be similar to our characterization of the W2164C variant, which causes hypertrophic cardiomyopathy (see 1.2, and overleaf) – we would need to generate genome-edited point mutations in mice and iPSC-CMs and characterize their functional and molecular impairment. The work on the characterization of the W2164C mutation has taken us >7 years, so the reviewer will appreciate that such *in vivo* characterizations are clearly beyond the scope of the current manuscript, while of great mechanistic interest for future work.

Figure Redacted

Figure Redacted

Reviewer 2

2.1 *Specify the charges on the phosphorylated residues and provide the reference for the Forcefields.*

We thank the reviewer for noting that we had omitted to mention in the Methods (though we had in the Results) that we have set the phosphorylated residues as each holding two negative unit charges, and that we had not referenced the phosAA19SB force-field. While in solution the protein will be able to access different charge states, which will likely be in rapidly exchanging equilibrium, we chose this fixed charge state as a proxy that maximises the possibility that salt-bridges and hydrogen bonds might be formed. We have added these details to the Methods [**Methods: Molecular Dynamics section**].

2.2 *Hydrogen bond occupancy should be directly stated (not just pictorially, but in numbers) in the important cases.*

We agree with the reviewer: while we had done so for some of the hydrogen bonds we describe in the manuscript, we were overly selective. We have addressed this in two ways: 1) we have included all the occupancy numbers for the residue pairs featured in Fig. 5 in the corresponding legend, and 2) we have produced a spreadsheet detailing the occupancy of all the bonds. The spreadsheet will be provided as Supplementary data, and deposited with all other data in a University of Oxford online repository (ORA-Data) entry that will accompany this work when published. [**Legend to Fig. 5, and Supplementary spreadsheet**].

2.3 *In the comparison between WT and pT2677 heterodimer, the authors state that the loss of the hydrogen bond (N107-G2674) is due to a twist in the loop in which it (pT2677) resides. However, the picture (Figure 5C) does not quite explain it. Rather it shows N107 to have moved away. Is it possible that pT2674 forms other interactions nearby that yanks away the loop? MD trajectories can clarify.*

We thank the reviewer for noting this mistake that stemmed from a legacy version of the initial submission. We agree with the reviewer that the change brought about by phosphorylation is better described differently. We have changed the description accordingly: “we noted that phosphorylation of T2677 leads to the loss of a hydrogen bond between the FLNC G2674 backbone carbonyl and the HSPB7 N107 sidechain” [**Results, p15**]

2.4 *Figure 6 is not explained adequately. For example, does it only show the WT? Are there corresponding plots for the variants or were they ignored because the interface distances were overlapping? In some cases, there were visible differences between the homodimer and heterodimer. So what are these residues, do they correspond to the residues involved in the hydrogen bonds shown in Figure 5?*

We have re-assessed and re-worked Fig. 6 in light of the reviewer’s comment [**Fig. 6, its legend, and Discussion p19**]. To answer the reviewer’s specific questions: the plots were only for the WT FLNC in the homo- and hetero-dimer; the point we are trying to convey with them is that these two forms of dimerization compete directly with each. We have remade the plot to also focus in more on the dimer interface, which allows the reader to locate phosphorylatable residues (as well as the others at the dimer interface, including ones picked out in Figure 5), and included the data for the phosphorylated forms. We think the figure is much improved as a result in how it relates to the results presented, and in enabling the discussion. For completeness we also include all these distances in the same spreadsheet as the hydrogen bond occupancy values [**Supplementary spreadsheet**].

2.5 *References for the SHAKE algorithm and Particle Mesh Ewald method are missing.*

We have inserted these references into the methods section [**References 67 and 68**].

2.6 *I am assuming that all the six systems simulated remained stable. There should be a statement about basic stability of the dimers.*

Yes, after the simulations remained stable over the 1- μ s production run of each of the six simulations. For each simulated system, the RMSD of the production run vs the protein structure pre-equilibration does

not show a systematic trend of increase over time (see Figure below, which omits the first frame). For homodimers, the RMSDs are always $<2 \text{ \AA}$ (with fluctuations $<0.5 \text{ \AA}$), whereas for heterodimers the RMSD is always $<2.9 \text{ \AA}$ (with fluctuations $<1.5 \text{ \AA}$). We have added a statement accordingly in the Methods section [Methods: *Molecular Dynamics* section], and added a version of the figure below into the Supplementary Information [new Supplementary Fig. 7A].

Reviewer 3

3.1 Whether T2677 and Y2683 are the only potential phosphorylation sites in this region in FLNC.

In our work we have mapped the dimer interface to the region spanned by residues 2665-2685 of FLNC. CST PhosphoSitePlus lists three post-translational modifications in this region (below), with only T2677 and Y2683 as phosphorylation sites at the dimer interface (red box). We have adjusted our Fig. 6 to focus on the dimer interface, allowing the reader to pinpoint these two phosphorylatable sites as well as others they consider relevant [Fig. 6].

Site	Sequence context	HTP
S2655-p	AFVGQkNsFTVDCSK ▼	1
T2677-p	VGVHGPKtPCEEV _y V ▼	1
Y2683-p	KtPCEEV _y VkHMGNR ▼	215
K2685-ac	PCEEV _y VkHMGNRVY ▼	1
K2699-sc	YNVTYTVkEKGD _y IL ▼	1
K2699-ac	YNVTYTVkEKGD _y IL ▼	1

<https://www.phosphosite.org/kinaseLibraryAction.action?siteId=1870675846> (accessed 23/07/24)

3.2 Whether the sequence reveals anything about potential kinases responsible for the phosphorylation.

To address this relevant question from the reviewer we have examined the results of prediction algorithms for kinases potentially responsible for both T2677 and Y2683 phosphorylation. As an example, the top five hits from PhosphoSite are shown below, for each site, and discussed separately.

kinase	kinase group	log ₂ (score)	site percentile	percentile rank
GSK3A	CMGC	6.864	99.988 %	1
GSK3B	CMGC	3.254	99.966 %	2
JNK1	CMGC	6.524	99.526 %	3
JNK3	CMGC	5.648	99.147 %	4
JNK2	CMGC	5.899	98.759 %	5

T2677: <https://www.phosphosite.org/kinaseLibraryAction.action?siteId=1870675846>

For T2677, both glycogen synthases kinase (GSK3s) and c-Jun N-terminal kinases (JNKs) are proposed. These kinases play important roles in cardiac pathologies related to biomechanical stress, e.g. inhibition of GSK3beta improves cardiac function in Arrhythmogenic cardiomyopathy⁴, a disease that can also be caused by genetic variants in *FLNC*⁵. In addition, JNKs are known to be upregulated upon biomechanical stress⁶.

Given that T2677 lies in a SP motif (proline-directed motif), there are also many other kinases including ERK and p38 that can phosphorylate in such a motif⁷. These kinases also have key roles in the (patho-)physiology of the heart^{8,9}. As such, and without extensive studies including *in vitro* kinase assays and cell-based inhibitor studies, it is not possible to pinpoint the kinases responsible at this stage.

kinase	kinase group	log ₂ (score)	site percentile	percentile rank
ITK	TEC	3.740	99.300 %	1
HER2	ErbB	2.864	99.263 %	2
ETK	TEC	3.864	99.116 %	3
FLT3	PDGFR	2.720	99.079 %	4
JAK2	JAK	2.189	99.043 %	5

Y2683: <https://www.phosphosite.org/kinaseLibraryAction.action?siteId=38509>

Regarding Y2683, for which a variety of tyrosine kinases are proposed, we note that the roles of tyrosine kinases in the heart are less well understood than in cancer. The best evidence for the importance of regulated tyrosine kinase activity comes from adverse cardiac effects of tyrosine kinase inhibitors used as cancer therapeutics¹⁰: systemic tyrosine kinase inhibition to treat cancer can lead to heart failure, arrhythmias and sudden cardiac death in cancer patients.

It is important to note that the experimental identification of kinases responsible for modulating a site *in vivo* is challenging. Most kinases show promiscuity in *in vitro* assays. One way to overcome this issue is to use complex kinase inhibitor profiling¹¹. Moreover, our own work¹² (and that of others) has shown that the temporal and special regulation of kinase isoforms can be crucial for mediating their biological functions in the heart. Given the complexity of identifying the kinases responsible for modification of T2677 and Y2683 *in vivo*, we refrain from speculating in the manuscript and instead identify this as an important part of the future work [Discussion, p21-22].

3.3 Are there any cardiac phospho-proteomic data sets that show either of these phosphorylation sites?

We agree with the reviewer as to the relevance of this question. Phosphorylation at T2677 has been reported in cardiac phospho-proteomics data sets previously¹³, where it was found to be upregulated in infarct hearts. Phosphorylation at Y2683, to the best of our knowledge, has not been. However, we would like to stress that the inclusion of modification at this site in our study was to a large part motivated by our wish to understand the dimer interface (the core focus of this paper), with this being an excellent site to perturb.

In preparing this revision, we have spent substantial time and effort to see if we could identify the phosphorylation of T2677 and Y2683 in cardiac samples from our laboratory. Specifically, we immunoprecipitated FLNC from cardiac lysates of MLP-KO hearts (one of our models of cardiac stress, Fig. 1), ran the precipitate on SDS-PAGE, excised the FLNC band, digested with trypsin, enriched the phosphopeptides with Fe-IMAC and TiO₂ before running on the mass spectrometer.

We can estimate that approximately 100 ng of FLNC was loaded onto the instrument. This was sufficient for us to be able to find pS2234, which is the most abundant phosphorylation site in FLNC (210 identifications in PhosphoSite, <https://www.phosphosite.org/siteAction.action?id=47945>), but in a different domain. However, this amount of protein was not sufficient sample to identify either pT2677 or pY2683 (below).

An estimate of sensitivity suggests that we would need significantly more material of FLNC protein to possibly detect pT2677 or pY2683. There are many potential reasons for this, including the lower abundance of these phosphosites compared with the pS2234 site, the loss of these sites during the FLNC-IP or subsequent sample preparation steps because they are less stable, or a lower ionization efficiency of the respective phosphopeptides in electrospray ionization mass spectrometry (ESI-MS), which would result in rather low intensity ions in MS scans that may not allow identification based on MS/MS scans. Unfortunately, it is not feasible for us to achieve the detection of these sites by further up-scaling of the experiments as it would require many more MLP-KO mouse hearts (approx. 50), which is not consistent with the responsible use of animals in biomedical research and would not be approved by the Ethical Review Committee.

To address the reviewer's query in the manuscript we have revised and bolstered the section describing the published evidence for pT2677 in cardiac phosphoproteomics. We also further emphasize that we chose to examine pY2683 (which has not yet been identified in cardiac tissue, to the best of our knowledge) as a counterpoint to pT2677 – i.e. a different dimer interface phosphorylation [**Results, p13**]. We also note in the discussion that we were not able (likely for the technical reasons described in detail above) to detect pT2677 in our mouse heart tissue, and stressed in general that our investigation of these phosphorylatable sites is to probe the structural insights we have obtained in our work, while also allowing other researchers to infer mechanistic impacts of phosphorylation on FLNC they might have observed in cell/tissue states under investigation [**Discussion, p21**].

Reviewer 4

4.1 Please provide the evidence that pT2677 is associated with cardiac stress. I note that it was identified in previous studies (refs. 39/40) but it would be useful to include the evidence of its upregulation under these conditions to support the bold statements being made in the manuscript title and abstract, and link the structural work presented here to the underlying biology. Likewise with pY2683.

The reviewer's comment is similar to one made by reviewer 3 (3.3). It would be nice to have specific evidence of upregulation in these conditions but we were not able to do so (see paragraphs below,

identical to in 3.3). As a result, we agree that both the title and abstract were perhaps too bold. We have adjusted them both (the title now reads “Filamin C dimerisation is regulated by HSPB7”), and consider them to be a fair description of the work presented [**Title, Abstract**].

Quoted from above: Phosphorylation at T2677 has been reported in cardiac phospho-proteomics data sets previously¹³, where it was found to be upregulated in infarct hearts. Phosphorylation at Y2683, to the best of our knowledge, has not been. However, we would like to stress that the inclusion of modification at this site in our study was to a large part motivated by our wish to understand the dimer interface (the core focus of this paper), with this being an excellent site to perturb.

In preparing this revision, we have spent substantial time and effort to see if we could identify the phosphorylation of T2677 and Y2683 in cardiac samples from our laboratory. Specifically, we immunoprecipitated FLNC from cardiac lysates of MLP-KO hearts (one of our models of cardiac stress, Fig. 1), ran the precipitate on SDS-PAGE, excised the FLNC band, digested with trypsin, enriched the phosphopeptides with Fe-IMAC and TiO₂ before running on the mass spectrometer.

We can estimate that approximately 100 ng of FLNC was loaded onto the instrument. This was sufficient for us to be able to find pS2234, which is the most abundant phosphorylation site in FLNC (210 identifications in PhosphoSite, <https://www.phosphosite.org/siteAction.action?id=47945>), but in a different domain. However, this amount of protein was not sufficient sample to identify either pT2677 or pY2683 (below).

An estimate of sensitivity suggests that we would need significantly more material of FLNC protein to possibly detect pT2677 or pY2683. There are many potential reasons for this, including the lower abundance of these phosphosites compared with the pS2234 site, the loss of these sites during the FLNC-IP or subsequent sample preparation steps because they are less stable, or a lower ionization efficiency of the respective phosphopeptides in electrospray ionization mass spectrometry (ESI-MS), which would result in rather low intensity ions in MS scans that may not allow identification based on MS/MS scans. Unfortunately, it is not feasible for us to achieve the detection of these sites by further up-scaling of the

experiments as it would require many more MLP-KO mouse hearts (approx. 50), which is not consistent with the responsible use of animals in biomedical research and would not be approved by the Ethical Review Committee.

An alternative strategy might have been to probe whether the kinases responsible for FLNC phosphorylation at T2677 and Y2683 are active in our mouse models, but this was not possible since they have not yet been identified (see 3.2).

To address the reviewer's query in the manuscript we have revised and bolstered the section describing the published evidence for pT2677 in cardiac phosphoproteomics. We also further emphasize that we chose to examine pY2683 (which has not yet been identified in cardiac tissue, to the best of our knowledge) as a counterpoint to pT2677 – i.e. a different dimer interface phosphorylation [Results, p13]. We also note in the discussion that we were not able (likely for the technical reasons described in detail above) to detect pT2677 in our mouse heart tissue, and stressed in general that our investigation of these phosphorylatable sites is to probe the structural insights we have obtained in our work, while also allowing other researchers to infer mechanistic impacts of phosphorylation on FLNC they might have observed in cell/tissue states under investigation [Discussion, p21].

4.2 Given that the authors are re-analysing published datasets, it would be useful to provide the data deposition (PDX?_ numbers) for reference.

We agree, and have added them [Supplementary Fig. 5, Legend]

4.3 While D/E are often used as phosphomimetics, it is possible to see different effects depending on which of these is used. Given their interpretation of the differential effects of phosphorylation of these two residues,

it is therefore important to check that this is not purely the result of the different amino acid substitutions that have been used for T2677 (D) and Y2683 (E).

The reviewer is correct that one needs to be careful that mutation to D/E is a reasonable mimic of phosphorylation. We chose to make the mutations T→D and Y→E to mimic pT and pY because these were the best match given the respective average alpha-carbon to oxygen distances (i.e. D<E, pT<pY from D: 3.3 Å, pT: 3.7 Å, E: 4.6 Å, pY: 7.7 Å; for two Cα-O distances in each case). We note this rationale for the choice in the text [Results, p13].

To address whether these mutations are good mimics of the actual phosphorylation we have performed four new MD simulations. We obtained 1-μs simulations on T2677D and Y2683E FLNC homo- and heterodimers, to compare with those for pT2677 and pY2683. We examined the 368 hydrogen bonds that were observed in the simulation of a least one FLNC form, and assessed the differences in occupancy between the phosphomimic and phosphorylated form for each (T2667D – pT2677, and Y2698 – pY2683).

Of the 736 hydrogen bond occupancy difference instances assessed from these simulations, 598 (81.2%) showed differences in occupancy lower than ±2%, while 57 (7.7%) exceeded ±10% difference, and 22 (3.0%) exceeded ±20% (below). This demonstrates that phosphomimicking and phosphorylation forms at sites 2677 and Y2683 have very similar impact on FLNC *in silico*, with no significant overall changes in structure between them.

Examining next the 28 hydrogen bond occupancy difference instances that involve sites 2677 and 2683 directly (as either the bond donor or acceptor), we find that 21 (75%) showed differences in occupancy lower than ±2%, while 5 (18%) exceeded ±10% difference, and 3 (11%) exceeded ±20%. This demonstrates that even in the direct contacts made at sites 2677 (whether by pT or D), or at 2683 (whether by pY or E), the occupancies are very similar.

We note that for 4 of the 5 hydrogen bonds that showed a difference in occupancy exceeding ±10%, that they were more occupied in the phosphorylated than phosphomimicking forms. This is consistent with 1) the Cα-O distances presented above: each phosphorylated residue is longer than its phosphomimicking equivalent (pT=3.7 Å > D=3.3 Å and pY=7.7 Å > E=4.6 Å) and therefore has longer “reach”; and 2) the phosphate groups (in pY and pT) carried two negative charges while the carboxylate groups (in D and E) only carried one. Both effects can act to make a particular bond stronger in the case of phosphorylation versus phosphomimicry.

In summary we can conclude that, given that producing homogeneous phosphorylated pT2677 and pY2683 protein is essentially unfeasible in the laboratory (we do not know the appropriate kinases, let alone their efficiency, and note the site-promiscuity of kinases in *in vitro* assays) phosphomimicry is not just a practical experimental alternative, but also a justified one. We are grateful to the reviewer for having prompted us to undertake these additional computational investigations and are pleased how they strengthen the work. We have added the histogram into the Supplementary Information, together with the above discussion [**new Supplementary Fig. 7B, and Results p14**] and have included all the occupancy values [**Supplementary spreadsheet**].

4.4 Can the authors given any suggestions as to what enzymes may be regulating these phosphorylation events which appear to play such crucial structural roles?

The reviewer's question echoes that made of reviewer 3. We are not able to identify the kinases responsible (see paragraphs below, identical to in **3.2**) at this stage.

Quoted from above: To address this relevant question from the reviewer we have examined the results of prediction algorithms for kinases potentially responsible for both T2677 and Y2683 phosphorylation. As an example, the top five hits from PhosphoSite are shown below, for each site, and discussed separately.

kinase	kinase group	log ₂ (score)	site percentile	percentile rank
GSK3A	CMGC	6.864	99.988 %	1
GSK3B	CMGC	3.254	99.966 %	2
JNK1	CMGC	6.524	99.526 %	3
JNK3	CMGC	5.648	99.147 %	4
JNK2	CMGC	5.899	98.759 %	5

T2677: <https://www.phosphosite.org/kinaseLibraryAction.action?siteId=1870675846>

For T2677, both glycogen synthases kinase (GSK3s) and c-Jun N-terminal kinases (JNKs) are proposed. These kinases play important roles in cardiac pathologies related to biomechanical stress, e.g. inhibition of GSK3beta improves cardiac function in Arrhythmogenic cardiomyopathy⁴, a disease that can also be caused by genetic variants in *FLNC*⁵. In addition, JNKs are known to be upregulated upon biomechanical stress⁶.

Given that T2677 lies in a SP motif (proline-directed motif), there are also many other kinases including ERK and p38 that can phosphorylate in such a motif⁷. These kinases have also crucial roles in the (patho-)physiology of the heart^{8,9}. As such, and without extensive studies including *in vitro* kinase assays and cell-based inhibitor studies, it is not possible to pinpoint the kinases responsible at this stage.

kinase	kinase group	log ₂ (score)	site percentile	percentile rank
ITK	TEC	3.740	99.300 %	1
HER2	ErbB	2.864	99.263 %	2
ETK	TEC	3.864	99.116 %	3
FLT3	PDGFR	2.720	99.079 %	4
JAK2	JAK	2.189	99.043 %	5

Y2683: <https://www.phosphosite.org/kinaseLibraryAction.action?siteId=38509>

Regarding Y2683, for which a variety of tyrosine kinases are proposed, we note that the roles of tyrosine kinases in the heart are less well understood than in cancer. The best evidence for the importance of regulated tyrosine kinase activity comes from adverse cardiac effects of tyrosine kinase inhibitors used as cancer therapeutics¹⁰: systemic tyrosine kinase inhibition to treat cancer can lead to heart failure, arrhythmias and sudden cardiac death in cancer patients.

It is important to note that the experimental identification of kinases responsible for modulating a site *in vivo* is challenging. Most kinases show promiscuity in *in vitro* assays. One way to overcome this issue is to use complex kinase inhibitor profiling¹¹. Moreover, our own work¹² (and that of others) has shown that the temporal and special regulation of kinase isoforms can be crucial for mediating their biological functions in the heart. Given the complexity of identifying the kinases responsible for modification of T2677 and Y2683 *in vivo*, we refrain from speculating in the manuscript and instead identify this as an important part of the future work [**Discussion, p21-22**].

4.5 Minor comments - Abstract needs to be read through carefully – there are some minor grammatical errors. ‘variety of important’ to ‘Variants are causative’

We thank the reviewer for spotting the mistake quoted, and have adjusted this. During our revision process we have also corrected any other grammatical errors we discovered.

References

1. Leber, Y. *et al.* Filamin C is a highly dynamic protein associated with fast repair of myofibrillar microdamage. *Hum Mol Genet* **25**, 2776–2788 (2016).
2. Wu, T. *et al.* HSPB7 is indispensable for heart development by modulating actin filament assembly. *Proc Natl Acad Sci U S A* **114**, 11956–11961 (2017).
3. Liao, W. C., Juo, L. Y., Shih, Y. L., Chen, Y. H. & Yan, Y. T. HSPB7 prevents cardiac conduction system defect through maintaining intercalated disc integrity. *PLoS Genet* **13**, e1006984 (2017).
4. Padrón-Barthe, L. *et al.* Severe Cardiac Dysfunction and Death Caused by Arrhythmogenic Right Ventricular Cardiomyopathy Type 5 Are Improved by Inhibition of Glycogen Synthase Kinase-3 β . *Circulation* **140**, 1188–1204 (2019).
5. Hall, C. L. *et al.* Filamin C variants are associated with a distinctive clinical and immunohistochemical arrhythmogenic cardiomyopathy phenotype. *Int J Cardiol* **307**, 101–108 (2020).
6. Wakatsuki, T., Schlessinger, J. & Elson, E. L. The biochemical response of the heart to hypertension and exercise. *Trends Biochem Sci* **29**, (2004).
7. Johnson, J. L. *et al.* An atlas of substrate specificities for the human serine/threonine kinome. *Nature* **613**, 759–766 (2023).
8. Gallo, S., Vitacolonna, A., Bonzano, A., Comoglio, P. & Crepaldi, T. ERK: A key player in the pathophysiology of cardiac hypertrophy. *Int J Mol Sci* **20**, (2019).
9. Marber, M. S., Rose, B. & Wang, Y. The p38 mitogen-activated protein kinase pathway-A potential target for intervention in infarction, hypertrophy, and heart failure. *J Mol Cell Cardiol* **51**, 485–490 (2011).

10. Shyam Sunder, S., Sharma, U. C. & Pokharel, S. Adverse effects of tyrosine kinase inhibitors in cancer therapy: pathophysiology, mechanisms and clinical management. *Signal Transduct Target Ther* **8**, (2023).
11. Watson, N. A. *et al.* Kinase inhibition profiles as a tool to identify kinases for specific phosphorylation sites. *Nat Commun* **11**, (2020).
12. Lange, S. *et al.* MLP and CARP are linked to chronic PKC α signalling in dilated cardiomyopathy. *Nat Commun* **7**, (2016).
13. Reitz, C. J. *et al.* Proteomics and phosphoproteomics of failing human left ventricle identifies dilated cardiomyopathy-associated phosphorylation of CTNNA3. *Proc Natl Acad Sci U S A* **120**, (2023).